# Condensation of LINE-1 is critical for retrotransposition

**Srinjoy Sil, Sarah Keegan, Farida Ettefa, Lance T Denes, Jef D Boeke, Liam J Holt***

Institute for Systems Genetics, New York University Langone Medical Center, New York, United States

**Abstract** LINE-1 (L1) is the only autonomously active retrotransposon in the human genome, and accounts for 17% of the human genome. The L1 mRNA encodes two proteins, ORF1p and ORF2p, both essential for retrotransposition. ORF2p has reverse transcriptase and endonuclease activities, while ORF1p is a homotrimeric RNA-binding protein with poorly understood function. Here, we show that condensation of ORF1p is critical for L1 retrotransposition. Using a combination of biochemical reconstitution and live-cell imaging, we demonstrate that electrostatic interactions and trimer conformational dynamics together tune the properties of ORF1p assemblies to allow for efficient L1 ribonucleoprotein (RNP) complex formation in cells. Furthermore, we relate the dynamics of ORF1p assembly and RNP condensate material properties to the ability to complete the entire retrotransposon life-cycle. Mutations that prevented ORF1p condensation led to loss of retrotransposition activity, while orthogonal restoration of coiled-coil conformational flexibility rescued both condensation and retrotransposition. Based on these observations, we propose that dynamic ORF1p oligomerization on L1 RNA drives the formation of an L1 RNP condensate that is essential for retrotransposition.

## Editor's evaluation

This valuable study describes a new system for tracking the formation of puncta by ORF1p, a nucleic acid binding protein encoded by the L1 retrotransposon, *in vivo*. The fact that RNPs form "membrane-less" structures is already established in other situations as the authors point out, but the work provides better-defined biochemical features, especially for RNA association and *in vivo* dynamics. Overall, the evidence for the conclusions is solid, and the work will be of interest to colleagues studying retrotransposition as well as biomolecular condensates.

*For correspondence:
Liam.Holt@nyulangone.org

## Introduction

Retrotransposons are genetic elements that replicate themselves within a host genome in a process known as retrotransposition by a 'copy and paste' mechanism that utilizes an RNA intermediate. The Long Interspersed Nuclear Element 1 (LINE-1 or L1) family of retrotransposons comprises 17% of the human genome by sequence and is the only autonomously active retrotransposon in the human genome, encoding proteins necessary for its own transposition. L1 also drives propagation of non-autonomous retrotransposons and processed pseudogenes, which make up an additional 21% of the human genome (*Lander et al., 2001*; *Jurka, 1997*; *Dewannieux et al., 2003*; *Kazazian and Moran, 2017*). While the majority of genomic human L1s have undergone truncations and mutations that have rendered them inactive, full-length, retrotransposition-competent L1s are 6 kilobases (kb) in length and have a 5′ untranslated region (UTR) containing a bidirectional promoter, two open reading frames (ORFs), ORF1 and ORF2, separated by a short inter-ORF linker, and a 3′ UTR with a weak

polyadenylation signal (*Figure 1A*; *Speek, 2001*; *Swergold, 1990*; *Dombroski et al., 1991*; *Doucet et al., 2015*; *Burns and Boeke, 2012*).

To undergo retrotransposition, the genomic L1 must first be transcribed by RNA polymerase II, polyadenylated, and exported into the cytoplasm where the two encoded proteins, ORF1 protein (ORF1p) and ORF2 protein (ORF2p), are translated. These proteins are both necessary for retrotransposition and exhibit *cis* preference, a phenomenon in which they are more likely to mobilize the mRNA from which they were expressed than a co-expressed L1 mRNA (*Wei et al., 2001*). While the mechanism for *cis* preference remains unclear, a prevalent model is that ORF1p and ORF2p bind to the mRNA from which they were translated cotranslationally or immediately after translation (*Boeke, 1997*). The two proteins are translated non-stoichiometrically, with a large excess of ORF1p, and co-assemble with L1 mRNA to form the L1 ribonucleoprotein (RNP), which is the functional unit of L1 (*Hohjoh and Singer, 1996*; *Taylor et al., 2013*). ORF2p has at least two crucial enzymatic activities required for retrotransposition, the reverse transcriptase (RT) required to produce dsDNA from the RNA template (*Feng et al., 1996*; *Cost et al., 2002*; *Mathias et al., 1991*) and the endonuclease (EN), which defines the target site in genomic DNA (*Feng et al., 1996*; *Cost et al., 2002*; *Mathias et al., 1991*). Once assembled, the L1 RNP must translocate to the nucleus where ORF2p uses its endonuclease activity to create a single-stranded nick in the DNA, and reverse transcribes the L1 mRNA into DNA. The ORF2p-encoded reverse transcriptase uses the free 3'-hydroxyl of the nicked DNA as a primer, directly synthesizing the L1 complementary DNA (cDNA) into the host genome in a mechanism called target-primed reverse transcription (TPRT) (*Feng et al., 1996*; *Cost et al., 2002*; *Mathias et al., 1991*; *Luan et al., 1993*).

ORF1p is a 40 kDa nucleic acid binding protein that assembles into a homotrimer, and has nucleic acid binding activity that is necessary for L1 retrotransposition (*Martin and Bushman, 2001*; *Martin et al., 2005*; *Kulpa and Moran, 2005*). ORF1p contains three structured domains, a coiled coil (CC) that mediates trimerization, an RNA recognition motif (RRM), and a C-terminal domain (CTD) that cooperates with the RRM to bind nucleic acids (*Figure 1A–B*; *Januszyk et al., 2007*; *Khazina and Weichenrieder, 2009*; *Khazina et al., 2011*). The N-terminal region (NTR) is the first 52 residues of ORF1p. This region is unstructured and contains phosphorylation sites and a basic charged patch that are necessary for retrotransposition (*Adney et al., 2019*; *Cook et al., 2015*; *Khazina and Weichenrieder, 2018*).

Many RNA-binding proteins participate in the formation of membraneless organelles, including nucleoli, stress granules, RNA processing bodies, and mRNA transport granules (*Brangwynne et al., 2009*; *Brangwynne et al., 2011*; *Feric et al., 2016*; *Zhang et al., 2015*; *Wheeler et al., 2016*; *Lee et al., 2020*). These biomolecular condensates demix from the surrounding cytoplasm or nucleoplasm and serve specific functions by selectively including proteins and nucleic acids based on their biochemical properties (*Hyman et al., 2014*). The process by which these membraneless compartments form is known as biomolecular condensation. One mechanism of condensation is phase separation, a density transition in which the constituent molecules exceed a critical concentration and separate into two coexisting phases, one with a higher density of the molecules of interest and one that is relatively depleted (*Flory, 1942*; *Huggins, 1942*). An alternative mechanism of condensation is gelation or percolation, which is a networking transition in which a dispersed solution of monomers and oligomers (a sol) switches into a system-spanning network (a gel) when a concentration threshold known as the percolation threshold is exceeded (*Broadbent and Hammersley, 1957*; *Harmon et al., 2017*). Recent conceptual work has proposed that phase separation and percolation can be coupled in certain cases, such that phase separation drives the formation of a dense phase that then undergoes percolation as the molecular concentration in the dense phase exceeds the percolation threshold (*Mittag and Pappu, 2022*). Phase separation coupled to percolation (PSCP) allows for the formation of dense phases with viscoelastic material properties at higher concentrations while also driving oligomerization and the emergence of pre-percolation clusters in subsaturated solutions (*Harmon et al., 2017*; *Kar et al., 2022*; *Mittag and Pappu, 2022*). Importantly, whether these condensed phases behave more like viscous liquids or elastic solids depends on the timescales on which they are observed/perturbed and the densities and lifetimes of the molecular interactions that serve as physical crosslinks (*Alshareedah et al., 2021a*; *Alshareedah et al., 2021b*; *Mittag and Pappu, 2022*).

Across the numerous characterized phase-separating RNA-binding proteins, shared molecular features such as multivalency, intrinsically disordered domains, and structured RNA binding have been

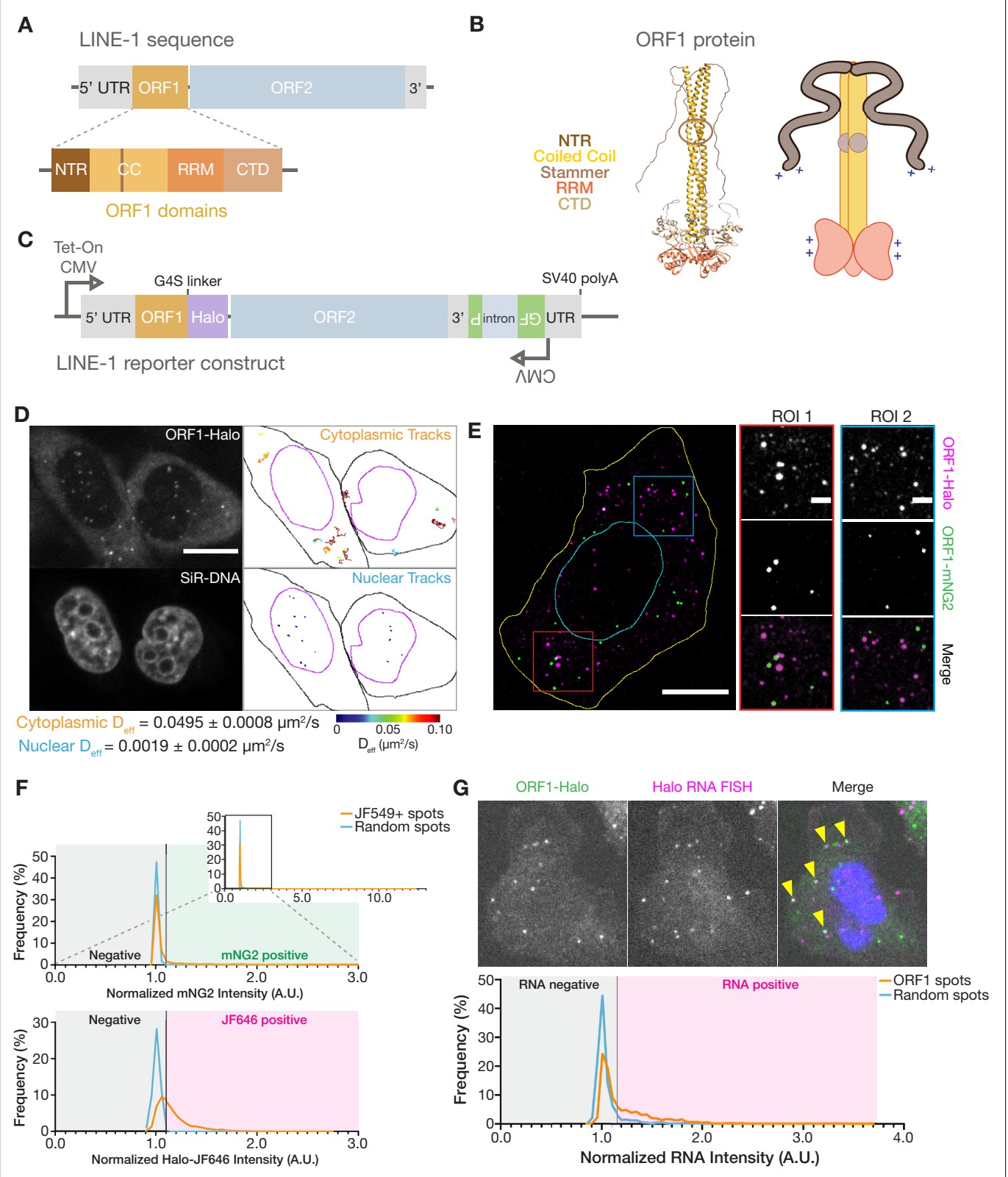

**Figure 1.** ORF1p forms monodisperse, diffusive puncta in live cells that do not readily mix. (**A**) Schematic of a full-length endogenous L1 element, with a detailed view of the domains of ORF1. (**B**) A structural model of an ORF1p trimer. A composite model generated by superimposing the coiled-coil structure (Protein Data Bank (PDB) entry 6FIA) on the RRM and CTD structure (PDB entry 2YKO). An extended conformation of the disordered NTR is modeled in. The flexibility-conferring stammer motif in the coiled coil is circled in brown. A cartoon model highlighting the motifs of interest

*Figure 1 continued on next page*

*Figure 1 continued*

is presented on the right. (**C**) Schematic of the modified L1RP element used for cellular expression. A Tet-On CMV promoter drives expression of the full-length L1, which also contains a C-terminal HaloTag (Halo) on ORF1p and a GFP-AI retrotransposition reporter in its 3' UTR. (**D**) ORF1 puncta diffuse much more slowly in the nucleus compared to the cytoplasm. A representative confocal micrograph at a single Z position shows ORF1 puncta in live HeLa cells after 24 hours of expression (top left) and corresponding nuclear staining (bottom left). Particle tracks generated from 10 s (100 frames) of imaging are shown for puncta in the cytoplasm (top right) and nucleus (bottom right) and are colored from blue to red, with blue indicating low effective diffusion and red representing high. Black and purple outlines represent hand-drawn cell and nuclear contours, respectively. Reported effective diffusion ($D_{eff}$) is the median and SEM $D_{eff}$ of 20 fields of view that each contain more than 5 puncta-containing HeLa cells following 24 hr of L1 expression. Scale bar = 5 µm. (**E**) ORF1p from co-expressed L1s predominantly condense separately in singly-labeled foci. Representative maximum Z projection image of a fixed HeLa cell expressing ORF1-Halo (magenta) and ORF1-mNG2 (green) off of two separate L1 expression constructs for 5 hr. Red and blue squares represent representative ROIs shown (right). Manually drawn lines reflect the contours of the cell (yellow) and nucleus (cyan). Scale bars = 10 µm (left) and 2 µm (ROIs). (**F**) ORF1p signal from co-expressed L1s colocalizes significantly less than a colocalization control. The cells were stained simultaneously with two Halo ligand dyes (JF549 and JF646), giving a positive control for colocalization. Histograms of mNG2 (top) and JF646 (bottom) intensity at JF549 + spots (orange) versus an equal number of random intracellular spots (blue) are shown. The full mNG2 histogram is shown (top-right) with an enlarged view to better visualize the histograms. The normalized intensity cutoff used to detect mNG2 and JF646 in the spots is shown and is the same for both histograms (Materials and methods). Mann-Whitney test between the JF549 + spot intensities and random spot intensities had a two-tailed p-value p<0.0001 for both mNG2 and JF646. N=1522 JF549 + spots and 1522 random intracellular spots across 31 cellular ROIs. (**G**) ORF1 puncta are frequently associated with L1 reporter RNA. Representative image of a fixed HeLa cell expressing the L1 reporter construct with fluorescently-labeled ORF1-Halo (green) and HCR-RNA FISH for the HaloTag sequence in the reporter RNA (magenta) (top). Nuclear staining is shown in blue in the merged image and yellow arrowheads indicate colocalized ORF1-Halo and reporter RNA. Histogram shows normalized RNA channel intensities at detected ORF1-Halo spots (orange) versus an equal number of random cytoplasmic spots per cellular ROI (blue) (bottom). The intensity cutoff used to call RNA-positive spots is shown (Materials and methods). Mann-Whitney test between the two sets of intensities had a two-tailed p-value p<0.0001. N=1756 ORF1 spots and 1756 random cytoplasmic spots across 26 cellular ROIs. NTR = N-terminal region, CC = coiled coil, RRM = RNA recognition motif, CTD = C-terminal domain.

The online version of this article includes the following source data and figure supplement(s) for figure 1:

**Source data 1.** Cytoplasmic and nuclear ORF1 puncta tracking data.

**Source data 2.** ORF1-ORF1 colocalization data.

**Source data 3.** ORF1-L1 reporter RNA colocalization data.

**Figure supplement 1.** Increased L1 expression primarily increases the number of ORF1 puncta, but longer expression leads to the formation of larger stress-granule-like assemblies.

**Figure 1-figure supplement 1-source data 1** . ORF1 puncta counting time course data.

**Figure 1-figure supplement 1-source data 2** . ORF1 puncta intensity time point data.

shown to be important for condensation (*Sanders et al., 2020*; *Yang et al., 2020*; *Molliex et al., 2015*). Given that ORF1p exhibits all three of these molecular features of phase-separating proteins, we hypothesized that ORF1p undergoes condensation to carry out its roles in L1 RNP formation and chaperoning of L1 machinery. Recent work has confirmed that purified ORF1p is able to form a liquid-like condensed phase *in vitro* and that a truncated protein containing only the NTR and CC is sufficient for condensation (*Newton et al., 2021*). Here, we demonstrate that ORF1p expressed from a full-length active L1 element rapidly forms cytoplasmic condensates in cells that exhibit *cis* preference. We found that structured and unstructured charged residues and coiled coil flexibility were all necessary for cellular condensation. Biochemical reconstitution experiments revealed that the material properties of the reconstituted ORF1p condensates and their response to differing protein-RNA stoichiometries predicted their propensity to assemble in cells. ORF1p condensation *in vivo* correlated with L1 retrotransposition activity: disruption of nucleic-acid binding motifs or a critical flexibility motif led to loss of condensation and abrogated retrotransposition, while orthogonal restoration of trimer dynamics rescued both condensation and retrotransposition. Together these results indicate that condensation is critical for L1 retrotransposition. We propose that the biochemical properties of ORF1p are tuned to efficiently nucleate L1 RNP condensates cotranslationally, while also allowing for dynamic interactions between fully assembled L1 RNPs and host proteins and nucleic acids.

## Results

### ORF1p forms punctate foci in live cells

Previous studies focusing on the ORF1 protein in mammalian cells and tissues have reported a variety of localizations (*Doucet et al., 2016*; *Goodier et al., 2004*; *Sharma et al., 2016*; *Goodier et al., 2007*; *Rodić et al., 2014*; *Pereira et al., 2018*; *Mita et al., 2018*). We used fluorescence microscopy to directly observe the behavior of the protein in live mammalian cells. We designed an inducible, active, endogenous-like L1 expression construct in which ORF1p was fused at its C-terminus to a HaloTag (ORF1-Halo; *Figure 1C*; *Los et al., 2008*). We were then able to use the fluorescent Janelia Fluor HaloTag Ligands JF549 and JF646 (Halo-JF549 and Halo-JF646) to visualize ORF1p in live cells (*Grimm et al., 2015*).

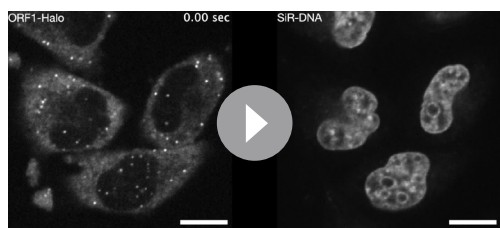

**Video 1.** ORF1 puncta diffuse more freely in the cytoplasm of HeLa cells than in the nucleus. A representative real-time confocal microscopy movie of ORF1-Halo puncta diffusing in the cytoplasm and nuclei of HeLa cells after 6 hr of L1 expression (left), with a corresponding confocal image of nuclear staining with SiR-DNA (right). Scale bars = 10 μm.
https://elifesciences.org/articles/82991/figures#video1

We also used a reporter to assess the ability of this engineered construct and its variants to complete the full L1 retrotransposition cycle. We used the well-characterized GFP-AI retrotransposition in the 3′ UTR of our construct (*Ostertag et al., 2000*; *An et al., 2011*; *Mita et al., 2018*). The GFP-AI cassette contains the EGFP coding sequence interrupted by the γ-globin intron. This intron is in the opposite orientation as the coding sequence and disrupts the GFP open reading frame. A CMV promoter and a thymidine kinase (TK) poly(A) signal flank the EGFP sequence, and the entire cassette is oriented antisense to the L1 sequence. When the L1 is transcribed, the γ-globin intron is removed by splicing, and successful retrotransposition of this spliced L1 mRNA construct allows for the subsequent expression of the uninterrupted EGFP coding sequence from a novel insertion site. This reporter enabled us to relate changes in ORF1p behavior upon mutation to the biological L1 retrotransposition function. Thus, we are able to directly relate the dynamics of ORF1p assembly to the ability to complete the entire retrotransposon life-cycle.

After 6 hr of L1 induction using the Tet-On system in HeLa cells, we noted the formation of bright, punctate, and highly uniform ORF1-Halo structures against a weaker background of diffuse ORF1-Halo signal (*Figure 1D*, *Figure 1—figure supplement 1A*). These foci were absent without induction, appeared as early as three hours following induction with doxycycline, and increased in number, but not size, with induction times up to 24 hr (*Figure 1—figure supplement 1A–B*). The ORF1p assemblies also increased slightly in fluorescence intensity with increasing induction times, indicating that this process is not pure phase separation with a fixed saturation concentration (*Figure 1—figure supplement 1C*; *Flory, 1942*; *Huggins, 1942*; *Alberti et al., 2019*). The ORF1 puncta that we describe in live cells are similar in size and uniformity to ORF1p assemblies seen using immunofluorescence in unstressed embryonal carcinoma cell lines (*Pereira et al., 2018*). Characterization of endogenous ORF1p in other cell lines has proven difficult due to the low L1 expression in somatic and non-transformed cells. Larger, stress-granule-like ORF1 condensates have been described in exogenous L1 expression systems and in cells that have been stressed with arsenite or thapsigargin treatment (*Goodier et al., 2007*; *Taylor et al., 2013*; *Mita et al., 2018*; *Pereira et al., 2018*). We observed similar stress-granule-like ORF1 condensates after inducing expression of our L1 construct for 72 hr (*Figure 1—figure supplement 1D*), suggesting that these larger assemblies are a result of very high ORF1p expression levels, while the monodisperse puncta seen after 6 hr may be more representative of ORF1p behavior at physiological expression levels. Although the diffuse ORF1-Halo signal remained predominantly cytoplasmic, nuclear ORF1-Halo puncta began to appear at longer induction times. Nuclear ORF1-Halo puncta had similar morphology and fluorescence intensity but very different diffusion behavior compared to cytoplasmic puncta (*Figure 1D*, *Figure 1—figure supplement 1E*): diffusivity was approximately 25 times higher for cytoplasmic puncta (*Video 1*). In summary, upon induction of L1 expression, ORF1-Halo quickly formed predominantly monodisperse puncta that diffused rapidly in the cytoplasm, whereas nuclear ORF1 puncta took longer to appear and exhibited lower diffusivity.

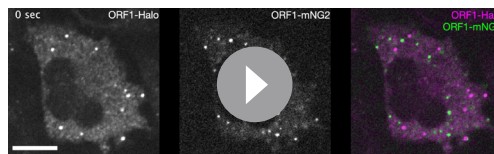

**Video 2.** Co-expressed ORF1 puncta exhibit minimal mixing in live HeLa cells. A representative movie of a HeLa cell co-expressing two L1s with ORF1 tagged with either HaloTag or mNeonGreen2 (mNG2) for 4 hr. Confocal images in each channel were acquired every 5 s for 1 min. Scale bar = 10 μm.

https://elifesciences.org/articles/82991/figures#video2

ORF1p and ORF2p are thought to exhibit preferential binding to the L1 mRNA from which they were translated, thus promoting transposition of intact and active L1s rather than mobilizing other L1-derived RNAs, host mRNAs, or competing retroelements such as Alu (*Boeke, 1997*; *Kaplan et al., 1985*). This phenomenon is known as *cis* preference, but its molecular mechanism remains unclear. There is functional evidence for the *cis* preference of ORF1p, as expression of an intact L1 only minimally complemented the retrotransposition of a co-expressed marked L1 encoding an ORF1 with an RNA-binding deficiency (*Wei et al., 2001*; *Kulpa and Moran, 2006*). Additionally, the L1s that were identified in cases of novel insertional mutagenesis were always derived from intact full-length L1s, suggesting that active L1s are far more likely to drive *cis* retrotransposition than they are to mobilize the much more abundant mutated or truncated L1s (*Kazazian et al., 1988*; *Woods-Samuels et al., 1989*; *Dombroski et al., 1991*; *Holmes et al., 1994*; *Moran et al., 1996*; *Brouha et al., 2003*; *Boeke, 1997*). We therefore wondered whether the ORF1p assemblies that we observed would form cotranslationally and stay separate, or conversely, could mix with each other, either directly through fusion events or indirectly through protein exchange with the surrounding cytoplasm.

To address this question, we simultaneously expressed two separate L1 elements differing only in the type of tag on the ORF1p (HaloTag or mNeonGreen2 (mNG2)) in the same cell. We used the Halo-JF549 to visualize ORF1-Halo (*Figure 1E*, magenta) and identified cells with expression of both ORF1-Halo and ORF1-mNG2 (*Figure 1E*, green). We almost always found singly-labeled puncta and only rarely observed puncta that contained both ORF1-Halo and ORF1-mNG2 (*Video 2*). As a positive control for our computational analysis of colocalization, cells co-expressing ORF1-Halo and ORF1-mNG2 were stained with Halo-JF646 in addition to Halo-JF549 prior to fixation. Each ORF1-Halo puncta contains many ORF1-Halo proteins and, therefore, stochastic incorporation of the two HaloDyes should lead to strong colocalization of Halo ligand signals. Indeed, we found that the JF646 signal at JF549 + puncta was significantly higher than the JF646 signal at an equal number of randomly selected intracellular spots of the same radius (p<0.0001). In contrast, the mNG2 signal at JF549 + spots was more similar to that of random intracellular spots (*Figure 1F*). However, there was occasional mNG2 signal at JF549 + spots: 21% of JF549 + had detectable mNG2, compared to 56% with detectable JF646 signal (detection defined as an intensity three standard deviations above the median intensity of random spots, Materials and methods). These results suggested that ORF1p assemblies do not rapidly exchange protein with the surrounding cytoplasm or undergo frequent fusion events and are perhaps instead kinetically trapped in an assembled form cotranslationally. This observation is consistent with previous work on the functional *cis* preference of L1-encoded proteins (*Wei et al., 2001*; *Kulpa and Moran, 2006*).

We next asked whether ORF1-Halo puncta contained *cis* L1 RNA. By staining for ORF1-Halo and performing hybridized chain reaction RNA fluorescence *in situ* hybridization (HCR-RNA FISH; *Choi et al., 2018*) for the HaloTag sequence specific to our exogenously introduced L1 in the same cells, we found that cytoplasmic ORF1 puncta were often enriched for *cis* L1 RNA signal, with 41% of ORF1 puncta being positive for RNA signal compared to 10% of random spots (*Figure 1G*, Materials and methods). This finding indicated that the ORF1 puncta visualized after 6 hr of induced L1 expression likely represented L1 RNPs. The absence of RNA signal in some ORF1 puncta could reflect a propensity for ORF1 puncta to fail to incorporate their *cis* RNA or could indicate a technical limitation of RNA FISH to efficiently label RNA inside ORF1 assemblies; for example, it may be difficult for the FISH probes to penetrate the ORF1p condensate. In this way, we showed that ORF1p puncta undergo surprisingly minimal mixing in live cells and incorporate *cis* RNA, suggesting a role for rapid ORF1p assembly in the *cis*-preference of L1 RNPs.

## Purified ORF1p forms liquid-like droplets and co-condenses with RNA *in vitro*

Next, we used *in vitro* biochemistry to determine whether a minimal reconstituted system could form ORF1 condensates. To this end, we purified full-length ORF1p from *E. coli*. We modified existing protocols (*Carter et al., 2020*) to maximize removal of protein-bound RNA and found that the protein purified as a homotrimer as previously described (*Figure 2—figure supplement 1*, Materials and methods). We fluorescently labeled the purified protein using an amine-reactive fluorescent dye to visualize protein distribution in our microscopy assays (*Nanda and Lorsch, 2014*). We used a ratio of at least 10:1 unlabeled:labeled ORF1p in our assays to minimize potential artifacts arising from modification of ORF1p with dye.

Purified ORF1p protein formed an extensive condensed phase at low micromolar protein concentrations in buffer with physiological pH and salt concentrations (*Figure 2A*). When the same concentration of protein was incubated in buffers with increasing salt concentrations, the mean intensity of the protein in the condensed phase, the partition coefficient of the protein, and the total condensed phase area all decreased, with no condensate formation observed above 300 mM KCl (*Figure 2D*, left). Decreasing protein concentration at a fixed salt concentration decreased the total condensed phase area but increased the protein partition coefficient. At 200 and 300 mM KCl, the range of protein concentrations tested spanned the phase boundary such that higher protein concentrations led to detectable condensate formation but lower protein concentrations did not. Taken together, these experiments demonstrated that purified ORF1p robustly condenses at physiological salt concentrations. Inhibition of condensation at higher salt concentrations suggests that electrostatic interactions play an important role in ORF1p assembly.

Since ORF1p has previously been described to have nucleic acid binding activity, we were curious to see how addition of RNA would affect ORF1p condensed phase formation. We generated a 2 kb fluorescently labeled RNA corresponding to the 5′ of the L1 mRNA for use in *in vitro* condensation assays. We used this 2 kb fragment because it was difficult to consistently generate full-length L1 RNA by *in vitro* transcription. It has been estimated that an RNA that is fully coated by ORF1p could be bound by one ORF1p trimer every 75 nucleotides (*Khazina et al., 2011*; *Taylor et al., 2013*). The full-length 6-kb L1 mRNA would therefore require ~80 ORF1p trimers or 240 ORF1p molecules to fully cover the L1 mRNA, and our 2 kb RNA would require 30 trimers or 90 ORF1p molecules, resulting in a predicted ORF1p:RNA ratio of 90:1. We decided to explore the effects of a large range of RNA stoichiometries on ORF1p droplet formation at a fixed protein concentration (*Figure 2B*). We noted that the ORF1p droplets became less spherical at 1,000:1 protein:RNA, with the dominant species appearing to be short chains of slowly fusing droplets. At 300:1 protein:RNA, the ORF1p condensed phase was primarily composed of a large network of branched fibrillar structures. These non-spherical structures that exist at higher relative RNA concentrations may reflect a combination of a change in the material properties of the droplets and the formation of new effective polymer interactions that lead to novel conformational restrictions (*Keenen et al., 2021*; *Seim et al., 2022*). The formation of these non-spherical structures at a protein:RNA stoichiometry lower than that predicted for a fully bound L1 RNA was surprising and appeared to be at odds with the diffusive ORF1 puncta observed in cells. However, our reconstitution approach mixed preformed ORF1p trimers with RNA, which is distinct from the cotranslational assembly process that is likely to occur in cells. The kinetics of these assembly processes are distinct and may lead to different architectures with differing resultant material properties. Additionally, the simple reconstituted system does not contain host factors [e.g. RNA helicases (*Tauber et al., 2020*; *Li et al., 2013*; *Goodier et al., 2012*)], nor does it recapitulate the complex, crowded intracellular environment, which can have dramatic effects on biomolecular condensation (*Delarue et al., 2018*). Nevertheless, despite the limitations of our reconstitution approach, we showed that the physical properties of ORF1p condensates are strongly impacted by RNA abundance, with increasing RNA concentrations leading to the formation of non-spherical structures rather than droplets.

We wondered how RNA addition would affect the propensity of ORF1p to condense in buffers with increasing salt concentrations. To enable comparison of the properties of the RNA-containing droplets with those containing protein alone, we used a low RNA stoichiometry (3,000:1 protein:RNA). This stoichiometry maintains a liquid-like condensed phase, thereby facilitating characterization by standard biophysical methods. When we mixed RNA with ORF1p at this stoichiometry, the labeled RNA was

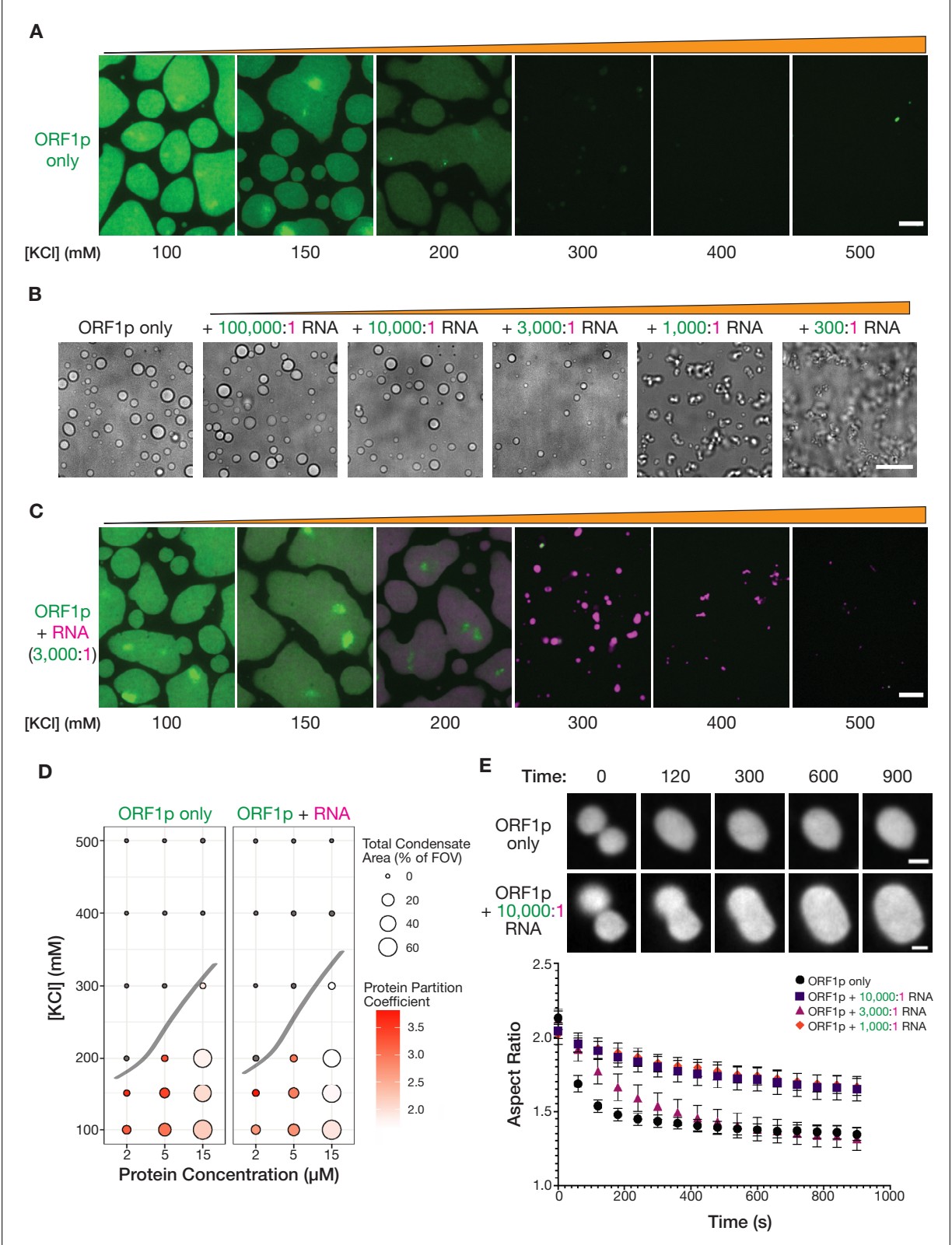

**Figure 2.** Purified ORF1p forms condensates with and without RNA, exhibiting differential condensate properties. (**A**) Purified ORF1p forms an extensive condensed phase *in vitro*. Representative images of ORF1p (green) condensed phase formation across a range of salt concentrations. 15 μM protein was used. All of the images use the same lookup tables (LUTs). Scale bars = 5 μm. (**B**) Increasing RNA leads to decreasing droplet area and eventually the formation of irregular three-dimensional fibrillar structures. Representative brightfield images of ORF1p droplet morphology over a wide range of

*Figure 2 continued on next page*

*Figure 2 continued*

2-kb L1 RNA stoichiometries, with increasing RNA concentration from left to right. 5 µM protein and 150 mM KCl was used in all conditions. Scale bar = 10 µm. (**C**) RNA robustly co-condenses with ORF1p *in vitro*. Representative images of ORF1p (green) condensed phase formation across a range of salt concentrations as in (**A**), in the presence of added labeled 2-kb L1 RNA (magenta). 15 µM protein and 5 nM RNA were used (3,000:1 protein:RNA). All of the images use the same LUTs for each channel. Scale bar = 5 µm. (**D**) RNA addition does not strongly affect the phase diagram of ORF1p *in vitro*. A phase diagram of ORF1p with and without added RNA (3,000:1 protein:RNA). Total condensed phase area of each condition is shown by the area of the circle for each condition, and the protein partition coefficient is represented by the filling. A hand-drawn phase boundary separates conditions with appreciable condensation with those that do not. (**E**) RNA-containing ORF1p condensates have slower droplet fusion kinetics than protein-only condensates. Representative images of fusion events over 15 min are shown for ORF1p with and without 10,000:1 protein:RNA addition, demonstrating slower fusion with RNA. 10 µM protein and 150 mM KCl were used in all conditions. Average aspect ratios across individual fusion events in each RNA condition are plotted (mean ± SEM) over time for 15 min. Ten or more fusions were analyzed per condition. Scale bars = 1 µm.

The online version of this article includes the following source data and figure supplement(s) for figure 2:

**Source data 1.** WT ORF1p *in vitro* droplet fusion data.

**Figure supplement 1.** Full-length ORF1p purifies as a trimer from a bacterial expression system.

**Figure 2-figure supplement 1-source data 1** . WT, K3A/K4A, and R261A ORF1p purification Coomassie gel images.

**Figure supplement 2.** ORF1p-RNA co-condensation suggests competing forces for protein and RNA partitioning.

robustly recruited to the ORF1p condensed phase (*Figure 2C*). Introduction of RNA into condensates at this stoichiometry did not strongly affect the total condensate area but did significantly decrease the protein partition coefficient compared to the corresponding condition without RNA, suggesting that RNA may compete with ORF1p protein-protein interactions for incorporation into condensates (*Figure 2D*, *Figure 2—figure supplement 2A–B*). Notably, we observed that the RNA partition coefficient increased with decreasing protein concentration and increasing salt concentration, indicating that more extensive protein condensation may limit RNA incorporation (*Figure 2—figure supplement 2C*). With these experiments, we showed that ORF1p co-condenses with RNA, with RNA incorporation leading to decreased protein partitioning, and increased protein concentrations leading to decreased RNA partitioning. These findings suggest that the protein-RNA interactions that promote RNA incorporation into ORF1p condensates are in competition with protein-protein interactions that drive ORF1p partitioning.

Taken together with our observation of non-spherical condensates at higher effective RNA concentrations, we wondered whether the RNA was altering the physical properties of the droplets. To assess the physical properties of the condensed phase, we analyzed droplet fusion events in time-lapse movies (*Alshareedah et al., 2021a*; *Video 3*). The dynamics of liquid droplet fusion are influenced by the surface tension and viscosity of the liquid as well as the size of the droplets (*Eggers et al., 1999*). We analyzed each fusion event over time by fitting an ellipse to the fusion intermediate at each time-point and calculating its aspect ratio. In a liquid-like fusion event, the aspect ratio will decrease exponentially to 1, which corresponds to a spherical fusion product. Analysis of other reconstituted protein droplets have shown that such fusions can occur on the order of a few seconds (*Elbaum-Garfinkle et al., 2015*). Droplets containing ORF1 protein alone exhibited slow fusion kinetics, requiring 3 min to reach a plateau of the aspect ratio (*Figure 2E*). Notably, the aspect ratio plateau was greater than 1, indicating that the fused droplets retained an ovoid shape. This could be explained by a high ratio of viscosity to surface tension, or could reflect a droplet maturation effect in which the droplets behave more like elastic solids rather than viscous liquids when observed over the course of minutes to hours (*Jawerth et al., 2020*; *Mittag and Pappu, 2022*). All RNA-containing ORF1p condensates fused more slowly than protein-only condensates, but surprisingly this decrease was non-monotonic with increasing RNA concentration: the addition of 10,000:1 and 1,000:1 protein:RNA resulted in fusions that plateaued at high aspect ratios, while 3,000:1 RNA condensates were able to fuse to similar final

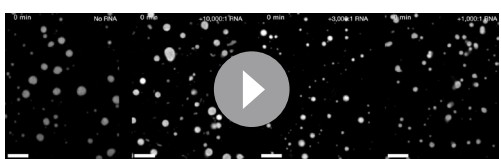

**Video 3.** ORF1p droplets exhibit slower fusion kinetics in the presence of RNA. Movies of ORF1p condensates settling out of solution and coalescing in the presence of varying amounts of RNA (from left to right: no RNA, +10,000:1 RNA, +3,000:1 RNA, and +1,000:1 RNA). 10 µM protein and 150 mM KCl were used in all conditions. Confocal images were acquired every minute for 2 hr. Scale bars = 5 µm.

https://elifesciences.org/articles/82991/figures#video3

aspect ratios as the ORF1p protein alone but took more time to reach the plateau (*Figure 2E*). These experiments indicated that addition of RNA to the ORF1p condensed phase changes its viscosity and surface tension in a way that slows droplet fusion kinetics. Additionally, they suggested that physical properties of the ORF1p condensed phase may change depending on the ratio of ORF1p to RNA, an effect that has been observed in other protein-RNA condensates (*Zhang et al., 2015*). Therefore, if cotranslational assembly drives L1 RNP assembly in cells, the physical properties of L1 condensates are likely to change during RNP assembly as ORF1p timers are sequentially added to the forming RNP, increasing the amount of ORF1p relative to the L1 RNA over time.

## Mutations of key basic residues alter ORF1p condensate properties *in vitro* and in cells

Electrostatic interactions modulated the properties of ORF1p droplets *in vitro*, implicating charged residues in the ORF1p condensation process. We were particularly interested in a positive charge patch at the end of the N-terminal disordered region, as these types of charged motifs have been implicated in nucleic acid binding and protein-protein interactions in other contexts, for example in the disordered tails of transcription factors (*Boija et al., 2018*; *Tóth-Petróczy et al., 2009*). Furthermore, two lysines within this charge patch, K3 and K4, have been previously reported to decrease L1 retrotransposition in cells when mutated (*Adney et al., 2019*; *Khazina et al., 2011*; *Khazina and Weichenrieder, 2018*). RNA interactions also affected the physical properties of ORF1p condensates in our *in vitro* experiments. Mutation of a central RNA-contacting arginine, R261, was previously shown to strongly decrease ORF1p RRM-mediated RNA binding *in vitro* and abrogated retrotransposition activity in cells (*Khazina et al., 2011*). Given these previous findings, we decided to investigate the condensation properties of K3A/K4A and R261A mutant proteins.

We purified full-length K3A/K4A and R261A mutant proteins. Both mutants were purified using the same protocol as wild-type (WT) ORF1p and were confirmed to be trimeric during size-exclusion chromatography (*Figure 3—figure supplement 1*), consistent with correct overall folding and assembly of the mutant proteins.

When reconstituted in buffer with physiological pH and salt concentration, both K3A/K4A and R261A ORF1p proteins formed condensed phases (*Figure 3A*). When assayed across a range of protein and salt concentrations, ORF1p K3A/K4A formed condensates in almost all of the same conditions as WT, while condensation of R261A was limited to only the conditions with higher protein concentrations and lower salt concentrations (*Figure 3B*). The decreased condensation of the R261A mutant was unexpected, as we predicted that mutating a core RNA-binding residue would only affect condensation in the presence of RNA. Notably, the total condensed-phase area of both mutants tended to be decreased compared to WT, as they formed smaller and more spherical droplets, with R261A having a decreased condensate area across all conditions in which it formed droplets (*Figure 3—figure supplement 2A*). We also noted that the protein partition coefficients of the R261A condensed phases were higher than their counterparts for WT and K3A/K4A, which further supported the hypothesis that R261 plays a role in driving ORF1p condensate assembly in the absence of RNA (*Figure 3B*, *Figure 3—figure supplement 2B*). Taken together, these experiments showed that K3/K4 and R261 are not essential for protein condensation *in vitro*, consistent with previous work indicating that neither the K3A/K4A mutation or deletion of the ORF1p RRM and CTD abrogated *in vitro* condensate formation (*Newton et al., 2021*). Mutations in K3/K4 and R261 instead limited condensate formation in conditions with low protein concentrations or high salt concentrations, implicating a role for these residues in protein-protein interactions that drive ORF1p condensation.

We next investigated the behavior of the mutant condensed phases in the presence of RNA. Both mutants robustly recruited the fluorescently labeled 2-kb L1 RNA into their condensates (*Figure 3C*). While the addition of RNA did not appreciably change the phase diagram of WT or K3A/K4A, the total area of R261A condensed phases increased with the addition of RNA, and co-condensation with RNA allowed R261A condensates to stably form at higher salt concentrations than protein alone (*Figure 3B–C*, *Figure 3—figure supplement 3A*). Additionally, the R261A condensed phases generally had higher RNA partition coefficients than WT or K3A/K4A (*Figure 3D*), despite the protein's reported deficiency in binding to structured RNA (*Khazina et al., 2011*). The recruitment of RNA to R261A condensates indicates that other parts of ORF1p are able to bind RNA with a moderate affinity, with likely areas including the disordered NTR and the N-terminal half of the coiled coil that

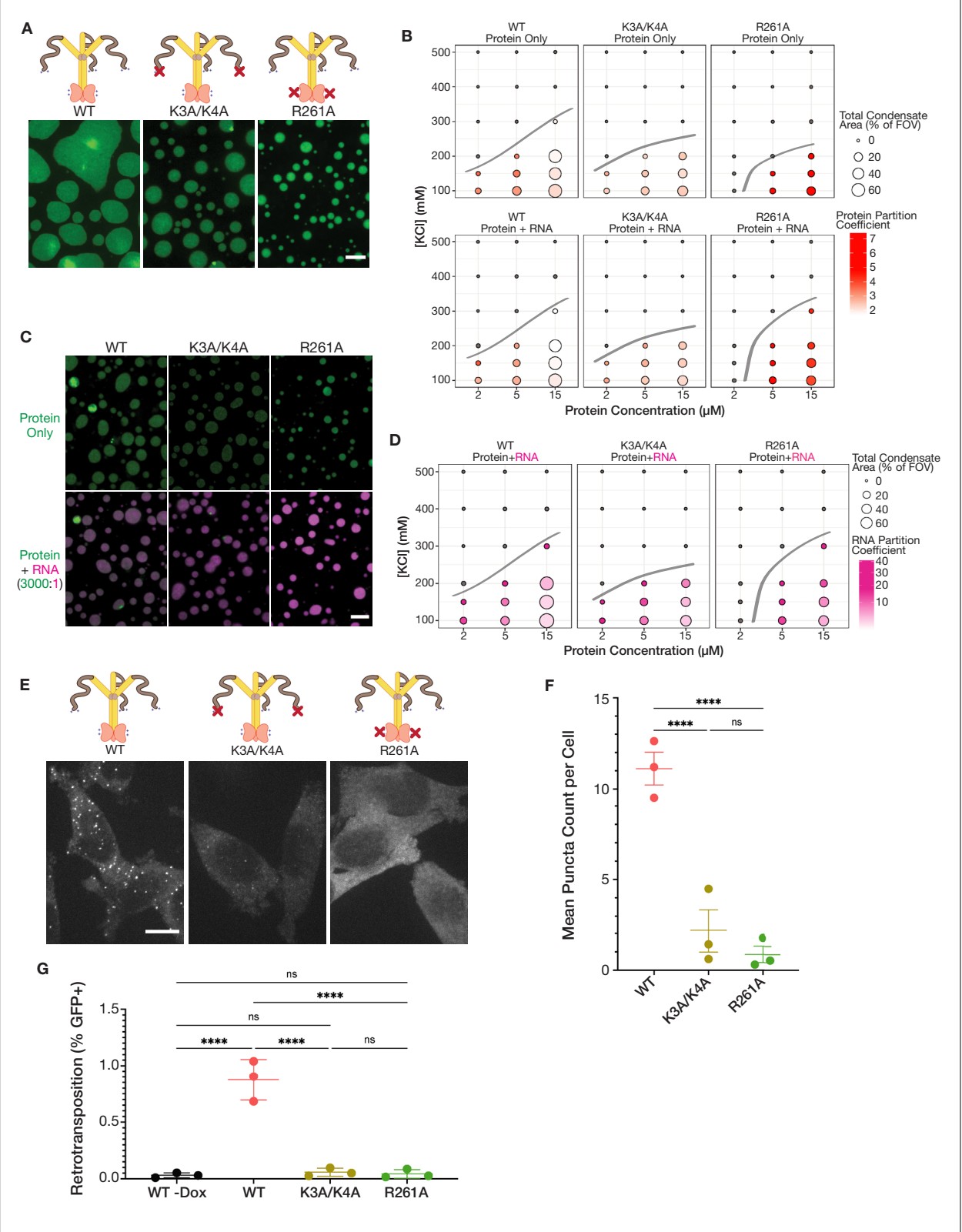

**Figure 3.** Mutations in key basic motifs attenuate droplet formation *in vitro* and abrogate condensation and retrotransposition in cells. (**A**) ORF1p variants with mutations in basic motifs form condensed phases *in vitro*. Representative images of the condensed phases of WT, K3A/K4A, and R261A ORF1p. 15 μM protein and 150 mM KCl were used for all three images. Scale bar = 5 μm. (**B**) Basic motif mutants of ORF1p have decreased propensity to form condensates in conditions with high-salt concentration or low protein concentration compared to WT. A phase diagram of ORF1p condensation

*Figure 3 continued on next page*

*Figure 3 continued*

for WT, K3A/K4A, and R261A with and without the addition of 3,000:1 2-kb L1 RNA, as in *Figure 2D*. R261A generates an RNA-responsive condensed phase with much higher protein partition coefficients than WT or K3A/K4A. Hand-drawn phase boundaries separate conditions with appreciable condensation with those that do not. (**C**) RNA enhances R261A condensation but has minimal effect on WT and K3A/K4A. Representative images of WT, K3A/K4A, and R261A ORF1p with and without RNA (3,000:1). All condensates were generated with 5 μM protein and 100 mM KCl. Protein image LUTs are the same per mutant as in (**A**), and RNA LUTs are the same across mutants. Scale bar = 5 μm. (**D**) ORF1p R261A tends to have higher RNA partition coefficients than WT and K3A/K4A. A phase diagram showing RNA partition coefficients for WT, K3A/K4A, and R261A ORF1p with RNA (3,000:1). Hand-drawn phase boundaries are included as in (**B**). (**E**) WT ORF1p forms cellular puncta much more robustly than K3A/K4A or R261A. Representative maximum intensity Z projections of HeLa cells expressing WT, K3A/K4A, or R261A ORF1p after 6 hr of L1 expression. All images have the same lookup tables. Scale bar = 10 μm. (**F**) Quantification of the average number of ORF1p puncta per cell after 6 hr of expression of WT, K3A/K4A, or R261A ORF1p. Each point represents one biological replicate of induction and quantification and is the average of at least 75 cells. The mean and SEM of three biological replicates are shown, and statistical differences between mutants were calculated using a one-way ANOVA with Tukey's multiple comparison correction. ****$p<0.0001$, ns = not significant. (**G**) WT L1 with ORF1-Halo undergoes retrotransposition at a cellular frequency of ~1%, while elements with ORF1 K3A/K4A and R261A have undetectable retrotransposition activity. Measured retrotransposition activity of WT, K3A/K4A, and R261A ORF1p after 72 hr of L1 expression. GFP+ cells were evaluated using FACS with a GFP+ threshold defined by WT cells without expression induction (WT -Dox). Each point is a biological replicate whose value is the average of three technical replicates, with 25,000 cells analyzed for each. The mean and SEM of three biological replicates are shown, and statistical differences between conditions were calculated using a one-way ANOVA with Tukey's multiple comparison correction. ****$p<0.0001$, ns = not significant.

The online version of this article includes the following source data and figure supplement(s) for figure 3:

**Source data 1.** WT, K3A/K4A, and R261A ORF1p cellular ORF1 puncta count data.

**Source data 2.** WT, K3A/K4A, and R261A L1 retrotransposition data.

**Figure supplement 1.** ORF1p K3A/K4A and R261A mutants purify as trimers.

**Figure supplement 2.** The condensed phases of ORF1p basic motif mutants are less extensive and have altered protein partitioning compared to wild-type.

**Figure supplement 3.** RNA addition has differential effects on the condensed phases of ORF1p basic motif mutants.

**Figure supplement 4.** ORF1p basic mutants are expressed at similar levels to WT but have no retrotransposition activity.

were not included in the ORF1p constructs that initially identified R261A as a mutant with loss of RNA binding (*Khazina et al., 2011*). Enhanced condensation of R261A but not K3A/K4A with RNA addition suggests that the contributions of K3/K4 and R261 to ORF1 protein condensation are mechanistically distinct and that the partial rescue of R261A condensation with RNA is dependent on K3 and K4. Indeed, the protein partition coefficients in matched conditions decreased with the addition of RNA for WT and R261A but not K3A/K4A (*Figure 3—figure supplement 3B*), providing evidence for N-terminal charge patch contributions to competing protein-protein and protein-RNA interactions. Overall, these experiments showed that ORF1p K3A/K4A and ORF1p R261A are able to co-condense with RNA and demonstrated that both K3/K4 and R261 contribute to distinct protein-protein and protein-RNA interactions that drive ORF1p condensation.

Given the altered condensation properties of these ORF1p mutants *in vitro*, we next sought to characterize their condensation in mammalian cells. After expressing WT, K3A/K4A, and R261A ORF1p in the context of a full-length L1 element for 6 hours in HeLa cells, we observed a stark reduction in puncta formation of the ORF1p mutants compared to WT. ORF1p K3A/K4A formed infrequent assemblies that appeared much smaller and dimmer than WT foci, while R261A staining was diffuse without any indication of condensate formation (*Figure 3E*). Quantification of the number puncta per cell confirmed the abrogation of bright puncta formation in both mutants (*Figure 3F*). Flow cytometry experiments showed that ORF1p protein expression was similar across cells expressing the WT and mutant L1s, indicating that loss of puncta formation was likely due to defective assembly rather than decreased protein abundance (*Figure 3—figure supplement 4A*). These experiments revealed that ORF1p K3A/K4A and R261A are not able to form WT-like assemblies when expressed in the context of full-length L1 in live cells, which contrasted with their mild condensation deficiencies in reconstitution experiments.

We wondered whether the loss of the ability to form bright ORF1p foci would have effects on L1 retrotransposition activity. A major advantage of our system is that these tagged ORF1p proteins were expressed in the context of an active L1 element with the GFP-AI retrotransposition reporter (*Ostertag et al., 2000*; *An et al., 2011*; *Mita et al., 2018*), which we used to assess the ability of each mutant to complete the entire L1 life-cycle. Using a well-characterized retrotransposition paradigm

that involves 72 hr of L1 expression, we found that our tagged WT element had readily detectable activity, undergoing retrotransposition in approximately 1% of cells (*Figure 3—figure supplement 4B*); this retrotransposition rate is consistent with previously reported rates for L1RP in similar constructs expressed in HeLa and HEK-293T cells (*Ostertag et al., 2000*; *An et al., 2011*). The K3A/K4A and R261A mutants, however, had undetectable retrotransposition activity (*Figure 3G*, *Figure 3—figure supplement 4C*). These findings demonstrated that both K3/K4 and R261 are necessary for ORF1p condensate formation in cells and retrotransposition, uncovering a possible connection between efficient ORF1p condensation and L1 retrotransposition activity.

## Dynamic coiled coils are necessary for ORF1p condensation

Coiled coils are common structural motifs consisting of superhelical bundles of α-helices that promote multimerization of proteins. Coiled coil sequences are formed from characteristic heptad repeats, in which a pattern of hydrophobic and hydrophilic amino acids repeats every seven residues (*Conway and Parry, 1991*). Insertions or deletions in this heptad repeat pattern have been shown to create local under- or overwinding of the supercoil that can affect coiled coil stability (*Brown et al., 1996*). Previous studies characterizing the ORF1p coiled coil showed that the C-terminal half of the coiled coil is sufficient for trimerization *in vitro* and has a higher evolutionary conservation than the N-terminal half of the coiled coil, which was found to undergo transitions between alpha-helical and unwound states rather than maintaining the tight three-fold symmetry of the C-terminal coiled coil (*Khazina et al., 2011*; *Khazina and Weichenrieder, 2018*). That work also characterized a three-residue stammer insertion (M91, E92, and L93) in the ORF1p coiled coil that, when deleted, increases the stability of the coiled-coil trimer and abrogates L1 retrotransposition (*Khazina and Weichenrieder, 2018*). The flexibility in the coiled coil was proposed to allow ORF1p trimers to interconvert between a closed state and an open state in which the N-terminus is capable of undergoing longer-range interactions that could facilitate inter-trimer interactions and network formation (*Khazina and Weichenrieder, 2018*).

As multivalency and dynamic interactions are key features of condensate-forming proteins, we hypothesized that stammer-deleted ORF1p would be deficient in condensation due to a decrease in inter-trimer interactions. Indeed, when we expressed L1 with a stammer-deleted (StammerDel) ORF1p in HeLa cells, we found that the StammerDel protein was unable to form punctate condensates (*Figure 4A*, left two panels). We reasoned that there the ORF1p stammer motif could lead play a role in ORF1p condensation through two non-exclusive mechanisms: (1) the chemical properties of the stammer's M, E, or L residues are necessary for condensation, or (2) the three-residue interruption in the heptad repeat pattern destabilizes the coiled coil leading to increased conformational dynamics that promote condensation. To test these hypothetical mechanisms, we generated two additional ORF1p mutants with orthogonal stammers, one with the stammer residues mutated to three alanines (StammerAAA) and one with the wild-type E92 restored in the tri-alanine stammer (StammerAEA). We chose to specifically investigate the role of E92 because it was shown to be the only stammer residue that is conserved across primate L1s (*Khazina and Weichenrieder, 2018*). Remarkably, both reconstituted stammer mutants formed puncta in cells (*Figure 4A*, right two panels). These experiments showed that destabilizing the coiled coil of ORF1p with a three-residue stammer insertion is necessary for condensate formation in cells.

## Dynamic coiled coils that drive ORF1p condensation are essential for L1 retrotransposition

We next sought to better characterize the behaviors of the stammer mutants in cells and how they might affect retrotransposition. Quantifying the average number of puncta per cell confirmed the StammerDel variant's inability to form condensates and revealed that StammerAAA formed a similar number of puncta per cell as WT, while StammerAEA formed about half as many puncta as WT (*Figure 4B*). We were surprised to find that restoring the wild-type E92 actually decreased ORF1p condensation; we note that E92 is adjacent to a methionine residue in all primate ORF1p sequences (*Khazina and Weichenrieder, 2018*), suggesting that the precise chemical properties of the stammer amino acids are finely tuned for optimal coiled-coil behavior. Flow cytometry confirmed that ORF1p expression in all three stammer mutant constructs was similar to WT, making it unlikely that these differences in condensation were due to protein abundance (*Figure 4—figure supplement 1A*). We then assayed the retrotransposition of these constructs and found that, while StammerDel was

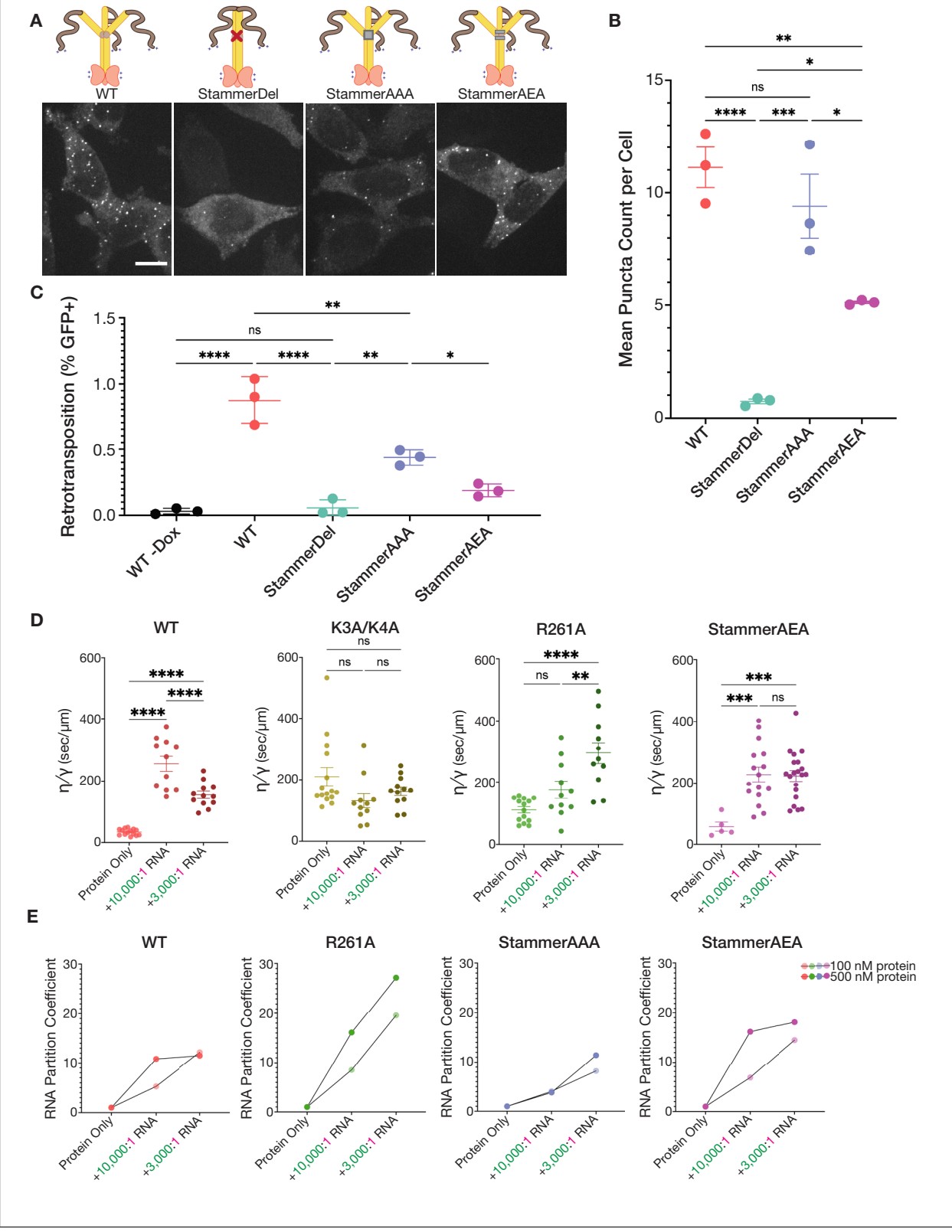

**Figure 4.** Stammer disruption of the ORF1p coiled coil is essential for L1 condensation in cells and modulates the physical properties of the ORF1p condensed phase. (**A**) Deletion of the stammer starkly decreases cellular ORF1p puncta formation, while stammer reconstitution rescues condensation. Representative maximum intensity Z projections of HeLa cells expressing WT, StammerDel, StammerAAA, or StammerAEA ORF1p after 6 hr of L1 expression. All images have the same lookup tables. Scale bar = 10 μm. ORF1p trimer cartoons displaying the corresponding mutations are shown,

*Figure 4 continued on next page*

*Figure 4 continued*

with hypothetical conformations based on the stammer's proposed role in promoting an open coiled-coil state. (**B**) Stammer reconstitution can rescue ORF1p puncta formation to WT levels. Quantification of the average number of ORF1p puncta per cell after 6 hr of expression of WT, StammerDel, StammerAAA, or StammerAEA ORF1p. Each point represents one biological replicate of induction and quantification and is the average of at least 75 cells. The mean and SEM of three biological replicates are shown, and statistical differences between mutants were calculated using a one-way ANOVA with Tukey's multiple comparison correction. *p<0.05, **p<0.01, ***p<0.001, ****p<0.0001, ns = not significant. (**C**) ORF1p stammer deletion abrogates retrotransposition, while stammer reconstitution rescues retrotransposition activity. Measured retrotransposition activity of WT, StammerDel, StammerAAA and StammerAEA ORF1p after 72 hr of L1 expression. GFP+ cells were evaluated using FACS as in *Figure 3G*. The mean and SEM of three biological replicates are shown, and statistical differences between conditions were calculated using a one-way ANOVA with Tukey's multiple comparison correction. *p<0.05, **p<0.01, ****p<0.0001, ns = not significant. (**D**) WT ORF1p and StammerAEA exhibit similar non-monotonic changes in the inverse capillary velocity ($\eta/\gamma$) of their condensed phases in response to increasing RNA concentrations. Inverse capillary velocity was calculated from individual droplet fusion events in each condition, with each point representing a single analyzed fusion event; see Materials and methods for details. Mean ± SEM is shown; 5 or more fusion events were analyzed per condition. Changes in inverse capillary velocity were assessed across RNA conditions for each mutant independently using a one-way ANOVA with Tukey's multiple comparison correction. **p<0.01, ***p<0.001, ****p<0.0001, ns = not significant. (**E**) WT and stammer-mutant ORF1p exhibit attenuated RNA partitioning at nanomolar protein concentrations compared to R261A. R261A has increased RNA partition coefficients with both increased RNA stoichiometry and protein concentrations. Each point represents the RNA partition coefficient from a full FOV of a single condition. Lines connect the values that use the same protein concentration. 150 mM KCl was used for all conditions.

The online version of this article includes the following source data and figure supplement(s) for figure 4:

**Source data 1.** StammerDel, StammerAAA, and StammerAEA cellular ORF1 puncta count data.

**Source data 2.** StammerDel, StammerAAA, and StammerAEA L1 retrotransposition data.

**Source data 3.** ORF1p mutant *in vitro* droplet fusion data.

**Figure supplement 1.** ORF1p stammer mutant proteins express at similar levels as WT but have variable retrotransposition rates.

**Figure supplement 2.** ORF1p StammerAAA and StammerAEA purify as trimers.

**Figure 4-figure supplement 2-source data 1** . StammerAAA and StammerAEA ORF1p purification Coomassie gels.

**Figure supplement 3.** Stammer-mutant ORF1p variants form limited condensed phases *in vitro*.

**Figure supplement 4.** The physical properties of the condensed phases of ORF1p variants have differential responses to RNA.

**Figure supplement 5.** Nanomolar ORF1p concentrations allow for punctate assembly formation *in vitro* in the presence and absence of RNA.

inactive, StammerAAA and StammerAEA both had readily detectable retrotransposition activity (50% of WT activity for AAA and 20% for AEA) (*Figure 4C*, *Figure 4—figure supplement 1B*). The pattern of retrotransposition activity strikingly mirrored the relative levels of puncta formation, with StammerAAA exhibiting a greater rescue of retrotransposition activity than StammerAEA, although both had less activity than WT. These findings demonstrated that synthetic stammer insertions are capable of restoring ORF1p condensation and enable retrotransposition activity to an extent that corresponds to their relative ability to form cellular puncta. Together, these experiments suggest that efficient ORF1p condensate formation is crucial for L1 retrotransposition.

## ORF1p variants that form puncta in cells exhibit specific changes in physical properties at increasing RNA stoichiometries *in vitro*

We purified full-length StammerAAA and StammerAEA ORF1p proteins with the aim of identifying *in vitro* condensed-phase behaviors that distinguish ORF1p variants that form puncta in cells from those that do not. Both stammer mutants were purified using the same protocol as WT. Both mutants eluted from size exclusion chromatography as trimers, indicating correct folding and assembly (*Figure 4—figure supplement 2*). We were unable to purify native StammerDel ORF1p. Previous characterization of the stammer-deleted coiled coil alone required purification from inclusion bodies using denaturation followed by refolding (*Khazina and Weichenrieder, 2018*); we felt that using denatured and refolded proteins in *in vitro* condensation assays would complicate interpretation of our results, therefore we did not characterize this protein. When we assayed the AAA and AEA stammer mutants for condensation in buffer with physiological pH and salt concentration, we found that they formed round droplets but had a reduced total condensed-phase area compared to WT and even R261A (*Figure 4—figure supplement 3A*, top). Adding very low concentrations of labeled 2-kb L1 RNA (10,000:1 RNA) at this protein concentration further reduced the condensate area of the stammer mutants, while the WT and R261A condensed phases were less affected (*Figure 4—figure supplement 3A–B*). These

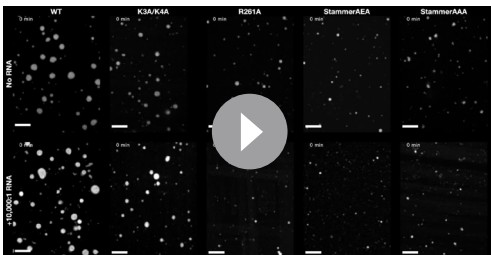

**Video 4.** Mutant ORF1p condensates exhibit differential changes in condensed phase material properties with the addition of RNA. Movies of ORF1p condensates settling out of solution and coalescing either without RNA (top) or in the presence of 10,000:1 RNA (bottom). All ORF1p variants (from left to right: WT, K3A/K4A, R261A, StammerAEA and StammerAAA) were assayed with 10 µM protein and 150 mM KCl. Confocal images were acquired every minute for 2 hr. Scale bars = 5 µm.

https://elifesciences.org/articles/82991/figures#video4

experiments suggested that the formation of an extensive condensed phase *in vitro* is not a strong predictor of the condensation of ORF1p variants in cells.

Testing a wider range of RNA concentrations further demonstrated that the stammer mutants undergo condensate morphology changes at lower RNA concentrations than the other ORF1p variants (*Figure 4—figure supplement 4A*). Both stammer mutants exhibited decreases in their limited condensed phase areas with the addition of very low concentrations of RNA (100,000:1–3,000:1 RNA), forming punctate condensates and irregular droplet fusion intermediates that contrasted with the spherical droplets that WT formed under these conditions. Additionally, the stammer mutants both formed branched fibrillar condensates at 1,000:1 RNA, while WT and R261A retained a morphology similar to chains of slowly fusing droplets at the same RNA concentration. Similar to the stammer mutants, the ORF1p K3A/K4A condensed phase also transitioned to a fibrillar morphology at 1,000:1 RNA, despite being unaffected by lower RNA concentrations, suggesting that the N-terminal lysine residues and the stammer properties both play a role in maintaining the dynamic polymer network interactions that are necessary for spherical ORF1p-RNA co-condensates. These findings showed that the ORF1p stammer mutants, which robustly condense in cells, exhibit attenuated *in vitro* condensate formation and altered condensed phase material properties compared to WT ORF1p and the basic motif mutants across a large range of RNA concentrations.

We then wondered whether the *in vitro* material properties of the stammer-mutant condensed phases, rather than their droplet areas and morphologies, are better predictors of their propensity to assemble into condensates in cells. We analyzed droplet fusion events for all mutants in the absence of RNA and in the presence of two low RNA concentrations, 10,000:1 and 3,000:1 RNA (*Video 4*). Notably, StammerAAA did not undergo a sufficient degree of condensation at either RNA concentration to evaluate fusion characteristics, and is therefore not included in this analysis. Measuring the aspect ratio of droplet fusion events over time showed that WT and StammerAEA exhibited faster fusion kinetics than the other mutants in the absence of RNA (*Figure 4—figure supplement 4B*, left). We had difficulty detecting fusion events for StammerAEA without RNA. Given the extremely sharp reduction in aspect ratio over time for this mutant, this is likely because most fusions occur too rapidly in this condition to be detected at our imaging rate of once per minute. However, with the addition of a very low concentration of RNA (10,000:1 RNA), both WT and StammerAEA exhibited much slower fusion kinetics than they did in the absence of RNA, while the behavior of other mutants was largely unchanged (*Figure 4—figure supplement 4B*, middle). At a slightly higher RNA concentration (3,000:1 RNA), the fusion kinetics of all mutants were similar, with the exception of R261A, which fused more slowly than the rest (*Figure 4—figure supplement 4B*, right).

Importantly, the kinetics of droplet fusion depend on both the sizes of the droplets and their physical properties. The characteristic fusion time $\tau$ of two simple Newtonian liquid droplets suspended in a lower viscosity solution is given by $\tau \approx (\eta/\gamma) \cdot \ell$, where $\eta$ is the droplet viscosity, $\gamma$ is the surface tension, and $\ell$ is the characteristic length scale, or size, of the droplets (*Eggers et al., 1999*; *Brangwynne et al., 2011*). The aspect ratio versus time plot of each analyzed fusion event fit well to an exponential decay function, allowing us to extract a characteristic fusion time $\tau$ for each fusion. We then divided each fusion event's $\tau$ by its characteristic length $\ell$ to determine the ratio of the condensed phase's viscosity to its surface tension ($\eta/\gamma$), a value that is also known as its inverse capillary velocity (*Brangwynne et al., 2011*; *Alshareedah et al., 2021a*). When we compared the inverse capillary velocity of WT ORF1p across the RNA conditions, we noted a non-monotonic effect. The WT inverse capillary velocity increased sharply with the addition of a very low concentration of RNA (10,000:1

RNA) and subsequently decreased at a slightly higher concentration of RNA (3,000:1 RNA) (*Figure 4D*, left). K3A/K4A and R261A had markedly different responses to RNA addition. K3A/K4A showed no appreciable changes in inverse capillary velocity in response to RNA (*Figure 4D*, middle left). R261A, in contrast to both, did not exhibit a change in inverse capillary velocity at the lower RNA concentration but had a substantial increase at the higher RNA concentration (*Figure 4D*, middle right). Significantly, StammerAEA showed a similar non-monotonic relationship between inverse capillary velocity and RNA stoichiometry as WT, exhibiting a steep increase at the lower RNA concentration and remaining stable at the higher RNA concentration (*Figure 4D*, right). Taken together, these experiments demonstrated that WT and StammerAEA ORF1p exhibit distinct non-monotonic changes in the inverse capillary velocity of their *in vitro* condensed phases in response to increasing RNA concentration, suggesting that differential condensed phase material properties in the presence of varied protein-RNA stoichiometries might be important for condensate formation in cells.

Since these droplet fusion experiments required micromolar concentrations of ORF1p in order to observe fusion events, we wondered if the differential effects of protein-RNA stoichiometry on the ORF1p variants could be observed at sub-micromolar protein concentrations. To that end, we assayed the condensation of ORF1p variants at high nanomolar concentrations, without RNA and in the presence of 10,000:1 or 3,000:1 RNA. Notably, all variants assayed formed punctate, sub-micron assemblies both in absence and presence of RNA, which appeared to be more consistent with the cellular ORF1 puncta than the previously observed droplets (*Figure 4—figure supplement 5A*). We observed that the ORF1p variants exhibited a differential propensity to partition RNA into condensates at the two RNA stoichiometries (*Figure 4E*). We noted that WT, StammerAAA, and, to a lesser extent, StammerAEA have an attenuated partitioning of RNA into ORF1p condensates with increasing protein and RNA concentrations, while R261A more strongly partitions RNA into condensates with increases in both protein and RNA. While all variants formed protein condensates that strongly colocalized with RNA, R261A was the only variant that had lower colocalization at the higher RNA stoichiometry, which was likely due to the formation of RNA foci with protein content that was below our limit of detection (*Figure 4—figure supplement 5A–B*). Notably, the RNA partitioning behaviors across ORF1p variants paralleled the trends observed for the material properties of ORF1p-RNA condensates across protein-RNA stoichiometries, suggesting that the differential fusion properties across variants are due to differences in protein and RNA partitioning at the tested stoichiometries. Taken together, these findings suggest that the biochemical properties of ORF1p are finely-tuned to balance protein-protein and protein-RNA interactions in a way that is critical for condensation of ORF1p in cells and downstream L1 retrotransposition.

## Discussion

Given the established importance of RNP formation for L1 retrotransposition and the recent appreciation of the contribution of dynamic interactions to the function of RNA-binding proteins, we investigated how the properties of ORF1p assemblies contribute to L1 retrotransposition. Although ORF1p's major role in retrotransposition is thought to involve the formation of an RNP with the L1 mRNA and ORF2p in the cytoplasm, the diversity of ORF1p mutations that abrogate retrotransposition suggests the possibility of additional roles, including mediating RNP interactions with host proteins and participation in the downstream steps of nuclear translocation, DNA search for integration sites, or reverse transcription and integration (*Martin et al., 2005*; *Kulpa and Moran, 2005*; *Khazina et al., 2011*; *Taylor et al., 2013*; *Khazina and Weichenrieder, 2018*; *Taylor et al., 2018*; *Adney et al., 2019*). Using a well-characterized L1 expression system and modifying it to allow for live-cell imaging of ORF1p is a significant first step towards relating the physical properties of higher order ORF1p assemblies to L1 function in cells. Expanding the use of this approach to investigate the role of ORF1p in cells and tissues with high endogenous L1 expression, including germline tissues and neoplastic tissues, will likely provide further insights into ORF1p interactions with host cell physiology (*Branciforte and Martin, 1994*; *Rodić et al., 2014*; *Ardeljan et al., 2017*).

High spatiotemporal resolution imaging of ORF1p in live cells allowed us to investigate longstanding hypotheses regarding L1 RNP assembly. In particular, the ability of L1-encoded machinery to specifically drive retrotransposition of L1 mRNA while minimizing transposition of off-target substrates was hypothesized to occur through cotranslational binding of ORF1p and ORF2p to their encoding L1 mRNA in *cis* (*Boeke, 1997*). This idea was substantiated with data demonstrating that two coexpressed

L1 constructs exhibit only minimal *trans* complementation when one has a loss-of-function mutation in ORF1p or ORF2p (*Wei et al., 2001*; *Kulpa and Moran, 2006*). Our work adds additional evidence supporting the *cis* preference model. The minimal mixing observed for co-expressed fluorescent ORF1p puncta suggests that the physical properties of these assemblies minimize fusion events and protein exchange with the surrounding cytoplasm, providing a biophysical mechanism for L1 *cis* preference (*Figure 1E–G*, *Figure 4D–E*). This type of kinetically arrested condensate has been described in simulations and has been proposed as a mechanism for the coexistence of many small condensates at low, endogenous concentrations, rather than the thermodynamic expectation that the condensates would coalesce into a single, large equilibrium-like assembly (*Ranganathan and Shakhnovich, 2020*; *Mittag and Pappu, 2022*). A mechanism involving reversible co-condensation of ORF1p together with the L1 RNA from which it was translated would allow for *cis* substrate preference without sequence-specific RNA binding (*Kolosha and Martin, 2003*). Recent work has provided cellular evidence that disrupting the spatiotemporal regulation of expression of an RNA-binding protein and the RNAs it condenses with can abrogate the formation of compositionally distinct condensates, emphasizing the importance of dynamical control of protein-RNA co-condensation in cells (*Lin et al., 2023*).

We propose that ORF1p condensation in cells occurs cotranslationally, creating a viscous condensate with the L1 mRNA. Since this condensation must occur with the L1 mRNA from which it was translated for successful propagation, it requires a protein that is exquisitely sensitive to very low concentrations of RNA, which we found to be a biochemical characteristic shared by the ORF1p variants that robustly formed punctate assemblies in cells (*Figure 4D*). Cotranslational condensation of ORF1p with its *cis* RNA would also require initial low concentrations of protein to favor protein-protein assembly while limiting protein-RNA condensation to avoid incorporating non-template RNA molecules in the L1 RNP, which we found to be another shared feature of the ORF1p variants that underwent condensation in cells (*Figure 4E*). *Cis*-preferential L1 RNP assembly driven by dynamical control of ORF1p co-condensation with its source RNA is an attractive model due to the necessary spatiotemporal association between the L1 RNA and ORF1p during translation, but further work is required to understand how this type of cotranslational assembly would function on a mechanistic level. This proposed role of ORF1p condensation in L1 RNP assembly would further imply that L1s expressing ORF1p defective in condensation may exhibit increased *trans* mobilization of off-target RNAs. Data supporting this hypothesis has been reported for retrotransposition of the mammalian SINE *Alu*, which was observed to transpose twice as efficiently when driven by an L1 with ORF1p deleted rather than an intact L1 (*Dewannieux et al., 2003*).

The behavior of ORF1p condensates in biochemical reconstitution experiments was distinct from the puncta properties observed in cells. In particular, the extensive droplet-like phase and the droplet fusion events seen in reconstituted ORF1p contrasted with the uniform punctate morphology and lack of mixing seen in cells. We found that reducing protein concentrations and titrating RNA concentrations allowed us to better approximate the punctate condensates seen in cells (*Figure 2B*, *Figure 4—figure supplement 5A*). However, our reconstitution system did not fully recapitulate our observations in cells, in which WT ORF1p assembled but K3A/K4A and R261A did not (*Figure 3B and F*, *Figure 4—figure supplement 5A*). Our simple biochemical reconstitution experiments involve equilibrium mixing of protein and RNA, allowing for nucleation, growth, and fusion of condensates in the absence of additional perturbations. While we posit that the discrepancies we observe are due to a dynamical control mechanism in cells that is not reflected in our reconstituted system (*Lin et al., 2023*), there are many other buffer parameters and non-equilibrium processes in cells that could play a role in modifying ORF1p assembly kinetics, including macromolecular crowding, ATP hydrolysis, and post-translational modifications (*Khan et al., 2018*; *Delarue et al., 2018*; *Brangwynne et al., 2011*; *Brangwynne, 2011*; *Nott et al., 2015*; *Aumiller and Keating, 2016*; *Carlson et al., 2020*). Given this array of cellular parameters that may affect condensation of proteins in cells, we strove to ensure that we could characterize the behavior and properties of ORF1p assemblies in a cellular expression system that allowed us to simultaneously assess protein function. The ORF1 mutants in this study indicate that ORF1p condensation is critical for L1 retrotransposition. We are now poised to further leverage this system to more precisely elucidate the emergent properties conferred to L1 RNPs as a result of condensation and how they contribute to ORF1p's essential roles in retrotransposition.

A recent study also described phase separation behavior of ORF1p in biochemical reconstitution experiments (*Newton et al., 2021*). That study characterized the formation of an ORF1p condensed

phase *in vitro* that is mediated by electrostatic interactions. The authors additionally demonstrated that a truncated ORF1p containing just the disordered NTR and coiled coil (residues 1–152; NTR-CC) is sufficient for condensation. This description of NTR-CC sufficiency for condensation in reconstitution experiments is in agreement with our characterization of R261A, which has greatly reduced structured RNA-binding activity (*Khazina et al., 2011*) but is still able to condense *in vitro* (*Figure 3A–B*). We predict that both ORF1p R261A and the truncated NTR-CC protein undergo condensation *in vitro* through a mechanism similar to complex coacervation, which requires only unstructured charge-charge interactions (*Aumiller and Keating, 2016*; *Boeynaems et al., 2019*). Our data suggest that the ORF1p RRM participates in protein-protein interactions that drive condensation in the absence of RNA, while the unstructured NTR plays a smaller role in protein-only condensation (*Figure 3A–C*, *Figure 3—figure supplement 2*). However, the NTR charge patch allows for stark modulation of the physical properties of the condensate in response to low stoichiometries of RNA (*Figure 4D*), which is critical for condensation in cells (*Figure 3E–F*). The RRM has lesser contributions to the physical properties of ORF1p-RNA condensates but appears to downregulate the incorporation of RNA into the co-condensates (*Figure 3—figure supplement 3*), and we find that the combination of protein-protein and protein-RNA interactions of both the NTR basic patch and the RRM drives the formation of a condensate that is highly sensitive to very low RNA concentrations (*Figure 4D–E*). Furthermore, the flexibility of the coiled coil that separates these two motifs appears to be critical to the roles of the NTR and RRM in condensation, since the StammerDel mutant that has a more rigid coiled coil (*Khazina and Weichenrieder, 2018*) exhibits starkly reduced assembly in cells (*Figure 4A–B*). In this way, we propose that two distinctive basic motifs and a flexible coiled-coil scaffold allow ORF1p to undergo rapid, dynamically controlled condensation with *cis* RNA within the complex environment of the cell.

The L1 system characterized in this work employs a uniquely powerful combination of biochemical reconstitution, live-cell imaging, and functional phenotyping in cells. *In vitro* reconstitution allows us to study the biophysical properties of condensates in a minimal and controllable system. We can use live-cell imaging to observe assembly in cells and identify factors and mutations that modulate condensate formation and behavior in the cellular milieu. A well-characterized retrotransposition assay enables us to correlate changes in cellular condensation with functional effects on the entire retrotransposon life-cycle. Further characterization of this system will reveal additional biophysical determinants of L1 retrotransposition, which will deepen our understanding of functional biomolecular condensation within the complex cellular environment.

# Materials and methods

## Key resources table

| Reagent type (species) or resource | Designation | Source or reference | Identifiers | Additional information |
|---|---|---|---|---|
| Gene (human) | LINE-1 retinitis pigmentosa (L1RP; LINE-1; L1) | 10.1093/hmg/8.8.1557; 10.1016 /j.cell.2013.10.021; 10.7554/eLife.30094 | | |
| Gene (synthetic) | HaloTag | Promega | G7711 | |
| Gene (synthetic) | EGFP antisense intron retrotransposition reporter (GFP-AI) | 10.1093/nar/28.6.1418 | | |
| Gene (synthetic) | mNeonGreen2 (mNG2) | 10.1038 /s41467-017-00494-8 | | |
| Strain, strain background (*Escherichia coli*) | BL21(DE3) | Sigma-Aldrich | 69450 | Used for recombinant protein expression |
| Strain, strain background (*Escherichia coli*) | DH10B | Thermo Fisher Scientific | EC0113 | Used for molecular cloning |
| Cell line (human) | HeLa rtTA2$^S$-M2 (HeLa M2) | 10.1093/nar/gkp108; 10.1016 /j.jmb.2006.10.009; 10.7554/eLife.30058 | RRID:CVCL_WN71 | Cells were routinely tested for mycoplasma and were negative. |
| Recombinant DNA reagent | L1 reporter construct with wild-type ORF1 (pLH2035; plasmid) | This paper | | L1 reporter construct with wild-type ORF1-GGGGS-HaloTag and GFP-AI engineered into the L1RP sequence driven by a Tet-On promoter on a pCEP-puro episomal plasmid backb one |

*Continued on next page*

*Continued*

| Reagent type (species) or resource | Designation | Source or reference | Identifiers | Additional information |
|---|---|---|---|---|
| Recombinant DNA reagent | L1 reporter construct with wild-type ORF1 tagged with mNeonGreen2 (pLH2060; plasmid) | This paper | | L1 reporter construct with wild-type ORF1-GGGGS-mNeonGreen2 engineered into the L1RP sequence driven by a Tet-On promoter on a pCEP-puro episomal plasmid backbone |
| Recombinant DNA reagent | L1 reporter construct with ORF1 K3A/K4A (pLH2042; plasmid) | This paper | | pLH2035 with ORF1 mutations K3A and K4A |
| Recombinant DNA reagent | L1 reporter construct with ORF1 R261A (pLH2043; plasmid) | This paper | | pLH2035 with ORF1 mutation R261A |
| Recombinant DNA reagent | L1 reporter construct with ORF1 StammerDel (pLH2040; plasmid) | This paper | | pLH2035 with deletion of residues M91, E92, and L93 in ORF1 |
| Recombinant DNA reagent | L1 reporter construct with ORF1 StammerAAA (pLH2041; plasmid) | This paper | | pLH2035 with ORF1 mutations M91A, E92A, and L93A |
| Recombinant DNA reagent | L1 reporter construct with ORF1 StammerAEA (pLH2046; plasmid) | This paper | | pLH2035 with ORF1 mutations M91A and L93A |
| Recombinant DNA reagent | Human ORF1p purification construct (pMT538; plasmid) | 10.1002/art.41054; 10.1016 /j. cell.2013.10.021 | | Full length synthetic human ORF1p from ORFeusHS with an N-terminal HIS6-TEV sequence in a pETM11 backbone such that cleavage leaves only an N-glycine scar |
| Recombinant DNA reagent | Human ORF1p K3A/K4A purification construct (pLH2075; plasmid) | This paper | | pMT538 with ORF1p mutations K3A and K4A |
| Recombinant DNA reagent | Human ORF1p R261A purification construct (pLH2076; plasmid) | This paper | | pMT538 with ORF1p mutation R261A |
| Recombinant DNA reagent | Human ORF1p StammerAAA purification construct (pLH2037; plasmid) | This paper | | pMT538 with ORF1 mutations M91A, E92A, and L93A |
| Recombinant DNA reagent | Human ORF1p StammerAEA purification construct (pLH2077; plasmid) | This paper | | pMT538 with ORF1 mutations M91A and L93A |
| Sequence-based reagent | T7_L1RP_F (oSS0133; forward primer for the amplicon used to generate IVT 2-kb L1 RNA) | This paper | PCR primers | TAATACGACTCACTATAGGG GCCGCTCTAGCCCTGGAAT |
| Sequence-based reagent | L1RP_R (oSS0121; reverse primer for the amplicon used to generate IVT 2-kb L1 RNA) | This paper | PCR primers | TGATTTTGCAGCGGCTGGTACC GGTTGTTCCTTTCCATGTTTAGCGCT |
| Commercial assay or kit | HaloTag Ligand JF549 | Promega | GA1111 | |
| Commercial assay or kit | HaloTag Ligand JF646 | Promega | GA1121 | |
| Commercial assay or kit | SiR-DNA | Cytoskeleton | CY-SC007 | |
| Chemical compound, drug | Hoechst 33342 | Thermo Fisher Scientific | 62249 | |
| Software, algorithm | NIS-Elements | Nikon | | |
| Software, algorithm | FlowJo | BD Biosciences | | |
| Software, algorithm | FIJI | 10.1038/nmeth.2019 | | |
| Software, algorithm | Prism 9 | GraphPad | | |
| Software, algorithm | RStudio | Posit | | |

## Plasmid construction

pLH2035 (pCEP-puro pTRE-Tight full-length L1RP containing ORF1-GGGGS-HaloTag and a GFP-AI cassette in its 3' UTR) was generated from a pCEP4 episomal plasmid vector in which the hygromycin resistance cassette was substituted with a puromycin resistance cassette and the CMV promoter was swapped with a pTRE-Tight Tet inducible promoter as previously described; an untagged full-length L1RP sequence was then ligated downstream of the promoter (*Taylor et al., 2013*). ORF1-HaloTag was made using Gibson assembly (*Gibson et al., 2009*) of a PCR DNA fragment encoding GGGGS-HaloTag from pPM285 (pCEP-puro-ORFeus ORF1-HaloTag, a gift from Jef Boeke) and pCEP-puro pTRE-Tight L1RP (*Mita et al., 2018*). The GFP-AI cassette was then digested and purified from pEA79 and ligated into the BstZ17I site in the 3' UTR of L1RP (*Mita et al., 2018*). Localized mutations in ORF1p (K3A/K4A, R261A, StammerDel, StammerAAA, and StammerAEA) were introduced into

pLH2035 using overlapping primers containing the mutation of interest to generate two PCR products that had homology to one another as well as to either a 5' NotI site or a 3' AfeI site that could then be Gibson assembled into the digested pCEP-puro-L1RP backbone, an approach similar to MISO mutagenesis (*Mitchell et al., 2013*).

pLH2060 (pCEP-puro pTRE-Tight full-length L1RP containing ORF1-GGGGS-mNeonGreen2) was generated from a pCEP-puro pTRE-Tight untagged L1RP construct. Gibson assembly was used to insert a GGGGS at the 3' end of ORF1 as well as an AscI site for fluorophore insertion. The mNeonGreen2 sequence was PCR amplified from pLenti6.2_mNeonGreen2 (a gift from Vanessa LaPointe; Addgene plasmid # 113727) and was inserted in the AscI tagging site using Gibson assembly.

pMT538 (pETM11-6xHis-TEV-hORF1p) was a generous gift from Martin Taylor and Kathleen Burns (*Carter et al., 2020*). Localized mutations in ORF1p (K3A/K4A, R261A, StammerAAA, and StammerAEA) were engineered using Gibson assembly of the digested pMT538 backbone with two overlapping PCR products containing the mutation of interest and homology to either the BamHI or XbaI sites, as previously described.

All bacterial transformations for molecular cloning were done in DH10B competent cells (Thermo Fisher Scientific, product number EC0113). Oligonucleotide primers for cloning were ordered from IDT unless otherwise specified. All constructs were verified by Sanger sequencing (Genewiz).

## Protein purification

Purification of full-length wild-type and mutant ORF1p was performed using a modified version of a previously described protocol (*Carter et al., 2020*). pETM11-6xHis-TEV-hORF1p constructs were transformed into BL21(DE3) competent *E. coli* cells (Sigma-Aldrich, product number 69450) using a standard bacterial transformation protocol and were selected on LB+kanamycin agar plates. A single colony was grown in a 20 mL overnight culture in LB+Kan (1 x LB with 50 µg/mL kanamycin) at 37 °C. The next day the cultures were diluted 1:100 in LB+Kan and the 2 L of culture were grown in a shaker until they reached OD 0.8. The cultures were then transferred to a 16 °C shaker for 1 hr, after which they were induced with 100 µM IPTG (EMD Millipore, product number 420322) overnight (18 hr) at 16 °C. The rest of the purification was done at 4 °C unless otherwise specified. The induced cultures were pelleted by spinning at 5,000 *g* for 10 min. The pelleted culture was resuspended in 40 mL of cold lysis buffer (50 mM HEPES + 500 mM NaCl + 25 mM Imidazole + 1 mM DTT + 1X EDTA-free protease inhibitor cocktail (Thermo Fisher Scientific, product number A32965) pH 8) and was lysed using sonication in an ice-water bath. Following sonication, the sample was incubated with 200 units of benzonase (Sigma-Aldrich, product number E1014) for 30 min. The crude lysate was cleared with two clearing spins at 14,000 *g* for 10 min. The cleared lysate was then incubated with 3 mL of 1:1 Ni-NTA agarose slurry (Qiagen, product number 30210) for 1 hr. Following pelleting, the Ni-NTA resin was washed twice with 10 mL of cold wash buffer (20 mM HEPES + 500 mM NaCl + 25 mM imidazole + 10 mM MgCl$_2$ + 0.1% Triton X-100 (Thermo Fisher Scientific, product number BP151-500) pH 8). An additional 5 mL wash was done using wash buffer containing 200 units of benzonase, 1 µg/mL RNase A (Thermo Fisher Scientific, product number EN0531), and 1 mM DTT for 3 hr. Three additional 10 mL washes were done with cold wash buffer, and then the His-tagged protein was eluted from the resin in 5 mL of cold elution buffer (20 mM HEPES + 500 mM NaCl + 500 mM imidazole + 10 mM MgCl$_2$ + 1 mM DTT pH 8) for 30 min. The eluted protein was then incubated with purified 6x-His-TEV protease E106G (*Cabrita et al., 2007*) and 500 µM TCEP (Thermo Fisher Scientific, product number 77720) for 20–24 hr. The cleavage mixture was then concentrated down to 500 µL in an Amicon-15 concentrator (EMD Millipore, product number UFC9050) and dialyzed back into lysis buffer overnight using small-volume dialysis chambers (Thermo Fisher Scientific, product number 88401). The protein mixture was subsequently incubated with 200 µL of Ni-NTA agarose slurry for 30 min to remove TEV protease as well as any uncleaved protein. The supernatant from this incubation wash then injected onto a Superose 6 Increase 10/300 GL size exclusion chromatography column (Cytiva, product number 29091596) on an AKTA pure protein purification system (Cytiva) and was run with cold, degassed gel filtration buffer (20 mM HEPES + 500 mM KCl pH 7.4). Fixed volume fractions were pooled based on A280 and A260 UV absorbance and ORF1p protein concentration was determined using NanoDrop A280 absorbance using a calculated molecular extinction coefficient of 25,440 M$^{-1}$cm$^{-1}$. The purified protein was then concentrated to 300 µM using an Amicon-2 Ultra concentrator (Millipore Sigma, product number UFC2030). DTT was added to the

concentrated protein to a final concentration of 1 mM, and the protein was aliquoted, flash frozen in liquid nitrogen, and stored at –80 °C.

For fluorescent labeling of purified ORF1p protein, approximately 10% of the purified protein was set aside prior to aliquoting and storage. Fluorescent ester dyes (green: Thermo Fisher Scientific, product number A37570; red: Thermo Fisher Scientific, product number A20003) were reconstituted in 10 µL of anhydrous DMSO (Thermo Fisher Scientific, product number D12345). The ORF1p protein was diluted to 500 µL in labeling buffer (20 mM HEPES + 500 mM KCl pH 6.5) and 1 µL of reconstituted dye was added to the solution and was mixed well by vortexing. The labeling mixture was then incubated mixing in the dark for 1 hr at 4 °C, after which it was set to dialyze into ORF-KCl buffer (20 mM HEPES + 500 mM KCl + 1 mM DTT pH 7.4) overnight at 4 °C. The protein concentration and moles dye per mole protein were calculated from NanoDrop absorbance measurements per manufacturer instructions. The protein was then aliquoted, flash frozen, and stored at –80 °C.

The final purified sample as well as purification intermediates were checked for purity on protein gels. Samples were banked in 4 X LDS Sample Buffer (Thermo Fisher Scientific, product number NP0007) with 5 mM DTT. Samples were boiled for 10 min at 95 °C, vortexed thoroughly, and centrifuged before loading onto a precast gel (Thermo Fisher Scientific, product number NP0329) and running with MOPS buffer (Thermo Fisher Scientific, product number NP0001). The gel was then stained using SimplyBlue SafeStain (Thermo Fisher Scientific, product number LC6060) per manufacturer guidelines, and gel images were acquired using the 700 nm channel of an Odyssey CLx scanner (LiCOR).

### *In vitro* transcription and RNA purification

A DNA template containing the T7 promoter and the L1RP 5′ UTR and ORF1 sequence was generated using PCR of pLH2035 with a forward primer containing the T7 promoter (T7_L1RP_F) and a reverse primer that bound near the 3′ end of ORF1 (L1RP_R). The 1970 bp DNA was then gel purified and 500 ng of DNA template was loaded into a 20 µL *in vitro* transcription reaction with 5,000 units of T7 RNA polymerase (New England BioLabs, product number M0251LVIAL), 1 X RNAPol Reaction Buffer (New England BioLabs, product number B9012SVIAL), 0.5 mM NTPs (New England BioLabs, product number N0450L), and 0.25 mM fluorescently labeled UTP. The labeled UTPs used were ChromaTide Alexa Fluor 488–5-UTP (Thermo Fisher Scientific, product number C11403), Cy3-UTP (Cytiva, product number PA53026), and Cy5-UTP (Cytiva, product number PA55026). The reaction mixture was incubated at 37 °C for 24 hr and subsequently underwent RNA clean-up using a column-based purification (Qiagen, product number 74104). The purified RNA was eluted in nuclease-free water (Qiagen, product number 120114) and was quantified using NanoDrop A260 absorbance as well as the Qubit RNA HS assay (Thermo Fisher Scientific, product number Q32852). RNA samples were diluted to 300 nM and were aliquoted, flash frozen, and stored at –80 °C.

### *In vitro* condensation assays

Prior to starting the condensation assays, the wells of a 384-well glass-bottom plate (Cellvis, product number P384-1.5H-N) were blocked as described in *Keenen et al., 2018*. Briefly, the wells were treated with 2% Hellmanex (Sigma-Aldrich, product number Z805939) for 1 hr, followed by three washes with ddH$_2$O. Wells were subsequently treated with 1 M sodium hydroxide for 30 min, washed three times with ddH$_2$O, and then incubated with freshly dissolved 20 mg/mL PEG-silane (Sigma-Aldrich, product number JKA3037) in 95% ethanol overnight at room temperature. The plate was parafilmed to prevent evaporation and was stored in the dark. The next day the PEG-silane solution was removed, the wells were washed three times with ddH$_2$O, and the wells were allowed to dry prior to plating protein mixtures.

ORF1p protein aliquots were thawed quickly at room temperature and were subsequently stored on ice, while fluorescently labeled RNA aliquots were thawed on ice. ORF1p unlabeled protein and labeled protein were mixed to a final concentration of 1–10% labeled protein and was subsequently diluted further with ORF1-KCl-Mg buffer (20 mM HEPES + 500 mM KCl + 1 mM MgCl$_2$ + 1 mM DTT pH 7.4), if necessary. The RNA sample was diluted with ORF1 No-Salt-Mg buffer (20 mM HEPES + 1 mM MgCl$_2$ + 1 mM DTT pH 7.4), if necessary. A total of 25 µL mixtures were made containing a mixture of ORF1-KCl-Mg buffer and ORF1 No-Salt-Mg buffer to adjust the final salt concentration of the mixture. The protein was always the final component to be added to the mixture, and after its

addition, the entire mixture was pipet mixed three times and plated immediately in a blocked well at room temperature. For end-point assays, the plated reactions were stored at room temperature in the dark for 2 hr prior to imaging. Fusion movies were imaged immediately following plating of all conditions, with images taken every minute. The plate was imaged on an Andor Yokogawa CSU-X confocal spinning disc on a Nikon Ti Eclipse microscope with 488 and 640 nm lasers (Coherent) used to excite the labeled protein and RNA, respectively. Brightfield and fluorescence images were recorded using a Prime 95B scMOS camera (Photometrics) with a 100 x objective (Plan Apo 100 x DIC, Nikon, oil, NA = 1.45, part number = MRD01905, pixel size: 0.09 µm).

End-point assay images were loaded in FIJI, and each field of view (FOV) was segmented into two regions (condensed phase and background) based on the 488 protein fluorescence signal, with the condensed phase intensity needing to be at least 2 x background intensity. Using the Analyze Particles function with a minimum particle size of 0.5 µm$^2$, this segmented image was then used to calculate total condensed phase area, mean condensed phase protein intensity, and mean condensed phase RNA intensity, as well as protein partition coefficient and RNA partition coefficient by taking the ratio between the mean protein or RNA intensity in the condensed phase and the mean protein or RNA intensity in the background area. Total droplet area and protein and RNA partition coefficients were plotted using R, filtering out partition coefficient values for conditions with less than 1% of the field of view occupied by condensed phase. For the nanomolar protein concentration experiments, the minimum particle size was decreased to 0.1 µm$^2$. For pairwise analyses between ORF1p variants or RNA stoichiometries, protein concentration-salt concentration conditions in which both variants/stoichiometries underwent measurable condensation were identified, and matched condensate values (total condensate area, protein/RNA partition coefficient) were compared using a paired two-tailed t test and plotted with lines connecting values from matched conditions. For colocalization, images underwent background subtraction, and the protein and RNA channels were analyzed for colocalization using the Coloc2 package in FIJI. Pearson's R value derived from the correlation of pixel intensities between the two channels was used as the colocalization statistic.

Droplet fusion movies were loaded in FIJI and underwent 3D drift correction. Individual droplet fusion events were extracted from each movie, with the criteria that the fusion had to occur primarily within the focal plane and was not affected by a third droplet during the 15 min fusion duration. Each fusion movie underwent segmentation to isolate the condensed phase, as previously described, and then the Analyze Particles function was used to fit an ellipse to the fusing droplets in order to determine the aspect ratio and the major and minor axis lengths of an ellipse fit to the fusing droplets in each frame of the 15 min fusion. The aspect ratio vs. time plot for each fusion event was fit to a One Phase Decay non-linear fit in Prism 9 (GraphPad) with the constraints K>0 and Plateau >1, allowing for the extraction of values for fusion time constant $\tau$ from each fusion event. Only fits with R$^2$ values greater than 0.95 were used for analysis. Average aspect ratio vs. time plots for the fusions from each mutant-RNA condition were generated in Prism 9. In order to calculate the inverse capillary velocity, or the ratio of viscosity to surface tension ($\eta/\gamma$), we used the equation $\tau \approx (\eta/\gamma) \cdot \ell$, dividing the fusion time constant $\tau$ by geometric mean fusion length $\ell = |(\ell_{major}(t=0) - \ell_{minor}(t=0)) \cdot \ell_{minor}(t=0)|^{1/2}$ (*Eggers et al., 1999*; *Brangwynne et al., 2011*). The inverse capillary velocity for each analyzed fusion event was then plotted per mutant per RNA condition and differences in the mean and distribution of values was compared across RNA conditions for each mutant using a one-way ANOVA with Tukey's multiple comparison correction.

## Mammalian cell culture

HeLa M2 cells [a gift from Gerald Schumann, Paul-Ehrlich-Institute (*Hampf and Gossen, 2007*; *Weidenfeld et al., 2009*)] were cultured in DMEM containing high glucose and sodium pyruvate (Thermo Fisher Scientific, product number 10313–039) supplemented with 10% FBS (Gemini, product number 100–106), 2 mM L-glutamine (Thermo Fisher Scientific, product number 25030–081), and 100 units/mL penicillin-streptomycin (Thermo Fisher Scientific, product number 15-140-122) (complete medium).

For transfection, HeLa M2 cells were seeded in a 6-well plate (Fisher Scientific, product number 50-202-137) on the day before transfection and were transfected with 1 µg of plasmid DNA per well using FuGENE-HD reagent (Promega, product number E2312) per manufacturer guidelines. HeLa M2 cells transfected with the L1 expression vector on a pCEP-puro episomal plasmid vector were

selected and maintained in complete medium supplemented with 1 µg/mL puromycin dihydrochloride (MedChem Express, product number HY-B1743A); the supplemented media with puromycin was sterile filtered with a 0.22 µm filtration device (Worldwide Medical Products, product number 51101007) prior to use (complete puromycin media). Transfected HeLa M2 cells were considered 'quasi-stable' for experimental use after at least 10 days of culture in complete puromycin media (*Mita et al., 2020*).

Cells were regularly split in fresh medium upon reaching 80–90% confluency. Cell culture medium was changed at least every 3 days. All cells were routinely tested for mycoplasma by PCR screening of conditioned medium.

## Mammalian live-cell imaging, puncta counting, and particle tracking

Quasi-stable HeLa cells with episomal inducible L1 expression constructs were seeded on non-coated 6-well, 12-well, or 96-well glass-bottom plates (Cellvis, product numbers P06-1.5H-N, P12-1.5H-N, and P96-1.5H-N) 1–2 days prior to imaging. Prior to imaging, the cells were induced with 1 µg/mL doxycycline hyclate (Sigma-Aldrich D9891) added to the conditioned media for the stated number of hours. The induced cells were then stained with 100 nM HaloTag Ligand JF549 (Promega, product number GA1111) and 500 nM SiR-DNA (Cytoskeleton, product number CY-SC007) in conditioned media per manufacturer instructions for 30–60 min. The cells were washed one time with DPBS (Fisher Scientific, product number 14-190-250) and fresh complete puromycin media was added to the cells for imaging.

Live cells were imaged on an Andor Yokogawa CSU-X confocal spinning disc on a Nikon Ti Eclipse microscope equipped with a stage top incubation system (Tokai Hit and World Precision Instruments) to maintain a temperature of 37 °C and 5% $CO_2$ in the recirculated air. The Halo and SiR-DNA signals were obtained using 561 and 640 nm lasers (Coherent), respectively, and fluorescence images were captured using a Prime 95B scMOS camera (Photometrics) with a 60 x objective (CFI Apo 60 x, Nikon, oil, NA = 1.49, part number = MRD01691, pixel size: 0.13 µm). 10 µm Z stacks of Halo and SiR-DNA fluorescence were taken for puncta counting, with 1 µm steps between frames. For puncta tracking, Halo signal was acquired at 10 frames-per-second (fps) for 10 s and a matched SiR-DNA image was acquired to differentiate nuclear and cytoplasmic puncta.

All live-cell images were analyzed in FIJI. For puncta enumeration, Z stacks were projected using maximum intensity and cells were manually outlined using Halo signal and were saved to the ROI Manager. Puncta in the Z-projected image were then identified using the Find Maxima function with a Prominence value of 75 and the number of maxima located within each cellular ROI was recorded. Cells with extremely high expression levels were excluded due to the inability to correctly identify puncta using the fixed prominence value, and mitotic cells were excluded since the Z stack did not include their full volume. Cells with zero maxima were included. The mean and SEM of the puncta count per cell for each induction condition or mutant was determined in Microsoft Excel (Microsoft). The average and SEM of the puncta count per cell for each induction condition or mutant was determined across three biological replicates using Prism 9 (GraphPad) and was plotted. The puncta count per cell values were compared across conditions or mutants using a one-way ANOVA with Tukey's multiple comparison correction where shown.

Puncta tracking was performed with the Mosaic suite of FIJI, using the following typical parameters: radius = 2, cutoff = 0, 20% of fluorescence intensity (Per/Abs), a link range of 1, and a maximum displacement of 5 px, assuming Brownian dynamics. Mean-square displacement (MSD) was then calculated for every 2D trajectory, selecting trajectories with more than 10 time points to reduce tracking error from particles moving in and out of the focal plane. We then fitted the time-averaged MSD of each selected trajectory with linear time dependence based on the first 10 time intervals (1 s): $MSD(\tau)_T = 4D_{eff}\tau$, where $\tau$ is the imaging time interval and $D_{eff}$ is the effective diffusion coefficient with the unit of $\mu m^2/s$, which ignores the effects of sub-diffusion and super-diffusion on individual tracks but allows for better comparison across different conditions. We used the median value of $D_{eff}$ among all trajectories within each field of view as a single data point and determined the mean and distribution of $D_{eff}$ values across all fields of view. This analysis was implemented in a Python software developed in our lab called GEMspa (*Shu et al., 2021*). Image areas corresponding to nuclei which contained tracked particles were manually outlined and a mask was generated that contained only those areas. The mask was inverted to create a mask containing all non-nuclear tracks, and these masks were used

to analyze nuclear tracks and cytoplasmic (non-nuclear) tracks separately. Particle intensities were calculated using the average pixel intensities for a 5x5 square around each particle's x,y position and were averaged across the length of the trajectory to generate a particle intensity for each track.

## ORF1 puncta colocalization assay, colocalization analysis, and fixed cell imaging

Quasi-stable HeLa M2 cells with the pLH2035 construct were transfected with pLH2060 in 6-well plates using Fugene-HD transfection reagent as previously described. The transfection mixture was replaced with complete puromycin media after 24 hr of incubation with the cells. The transfected cells were then seeded on non-coated glass-bottom 6-well plates (Cellvis, product number P6-1.5H-N) for 2 days. The cells were induced with doxycycline for 5 hr as previously described, and were subsequently stained with 100 nM Halo Ligand JF549 (Promega, product number GA1111) and 100 nM Halo Ligand JF646 (Promega, product number GA1121) for 1 hr in conditioned media containing doxycycline. The cells were then washed once with DPBS and were fixed in freshly diluted 4% formalin (Sigma-Aldrich, product number HT5012) for 10 min at room temperature. The cells were subsequently washed once with a PBS-Glycine solution (1 X PBS + 10 mM glycine + 0.02% sodium azide + 0.2% Triton X-100 [Thermo Fisher Scientific, product number BP151-500]) and twice with 1 X PBS. The cells were stained with 1 μM Hoechst 33342 (Thermo Fisher Scientific 62249) for 1 hr at room temperature in the dark and were subsequently stored in PBS at 4 °C in the dark until imaging. The fixed plates of cells were imaged on an Andor Yokogawa CSU-X confocal spinning disc on a Nikon Ti Eclipse microscope at room temperature. The fluorescence signals were obtained using DAPI epifluorescence excitation and the 488, 561, and 640 nm lasers (Coherent), and images were captured using a Prime 95B scMOS camera (Photometrics) with a 100 x objective (Plan Apo 100 x DIC, Nikon, oil, NA = 1.45, part number = MRD01905, pixel size: 0.09 μm). 8 μm Z stacks of Halo and SiR-DNA fluorescence were taken for puncta counting, with 1 μm steps between frames.

The images were analyzed in 3D space using a custom python script. Briefly, JF549+ spot positions on the 3D images were detected with the Laplacian of Gaussian method using the python package scikit-image (skimage.feature.blob_log). This function returns spot positions in the x, y and z planes, as well as spot radii. Quality checking was performed manually and input parameters for this function were adjusted based on the background level of an image. Spot positions were assigned to individual cell ROIs that had been manually drawn in ImageJ based on their x and y positions and spots outside of ROIs were excluded from the analysis. The mean intensity signal over a circle with matching radius and (x, y) position at a single z-level for each spot detected was measured and recorded for each channel of the image (JF646 and mNG2). To randomize spots, a count of spots was obtained per ROI and an equivalent number of random positions were selected from the pixel coordinates of each ROI. Since the ROI was drawn in two dimensions, but the spot localizations were in 3D, the z-channel position for each randomly chosen 2D position within an ROI was selected uniformly from the distribution of z-channel positions for the detected spots. The random spot radii were chosen in a similar manner. The intensities at a given spot were normalized within each ROI by dividing the given channel intensities by the median channel intensity of the random spots in the same ROI. The distributions of JF646 and mNG2 normalized intensities at JF549+ spots versus random spots were compared by Mann-Whitney tests in Prism 9 (GraphPad). JF646 and mNG2 were considered detected at a spot if the normalized intensity in the given channel was greater than 1.1. This threshold value was chosen as the normalized intensity value three standard deviations above the median intensity of the random spots in both the mNG2 and JF646 channels; the threshold value of 1.1 corresponds to a greater than 99th percentile value for mNG2 and JF646 intensity at random spots.

For qualitative imaging of ORF1-Halo in fixed cells, quasi-stable HeLa cells with episomal inducible L1 expression constructs were seeded in triplicate in glass-bottom 96-well plates (Cellvis, product number P96-1.5H-N) in complete puromycin media containing 1 μg/mL doxycycline for 72 hr. Cells were then stained with 100 nM Halo Ligand JF549 for 30 min in conditioned media containing doxycycline and washed once with DPBS. The cells were fixed as described above, using 4% formalin with washes with PBS-Glycine solution and PBS. Cells were stained with 100 nM SiR-DNA (Cytoskeleton, product number CY-SC007) for 1 hr at room temperature and were stored in the dark at 4 °C until imaging. The fixed plates of cells were imaged on an Andor Yokogawa CSU-X confocal spinning disc on a Nikon Ti Eclipse microscope at room temperature. The fluorescence signals were obtained using

the 561 and 640 nm lasers (Coherent), and images were captured using a Prime 95B scMOS camera (Photometrics) with a 40 x air objective (Plan Fluor 40 x DIC, Nikon, air, NA = 0.75, part number = MRH00401, pixel size: 0.275 µm). 5x5 large images were acquired with 20% overlap using the ND Acquisition feature NIS Elements software (Nikon) and were stitched together using the SiR-DNA channel.

## Hybridized chain reaction RNA fluorescence *in situ* hybridization (HCR RNA-FISH)

Probes were designed by and ordered from Molecular Instruments against the HaloTag coding sequence from pLH2035. FISH was performed according to the manufacturer's recommendation (*Choi et al., 2018*). Quasi-stable HeLa M2 cells transfected with the pLH2035 construct were seeded on non-coated glass-bottom 6-well plates (Cellvis, product number P6-1.5H-N) for 2 days. The cells were induced with doxycycline for 6 hr as previously described and were subsequently stained with 100 nM Halo Ligand JF549 (Promega, product number GA1111) for 1 hour in conditioned media containing doxycycline. The cells were then washed once with 1 X PBS (Thermo Fisher Scientific, product number 10010023) and were fixed in freshly diluted 4% PFA (Thermo Fisher Scientific, product number 50-980-495) for 10 min. The cells were subsequently washed three times with 1 X PBS, then permeabilized with 1% Triton X-100 (Fisher Scientific, product number BP151-500) in PBS for 10 min at room temperature. The samples were pre-hybridized with hybridization buffer (Molecular Instruments) that had been pre-warmed to 37 °C and placed at 37 °C for 30 min. Probes were diluted to 1 nM in pre-warmed hybridization buffer. Samples were incubated for 1 hr in the probe solution in a humidified chamber at 37 °C, then washed four times for 10 min each in pre-warmed wash buffer (Molecular Instruments) at 37 °C. Samples were then washed twice for 5 min in 5 X SSCT (5 X sodium saline citrate (SSC) with 0.1% Tween 20). Samples were then incubated in 200 µL amplification buffer (Molecular Instruments) for 30 min at room temperature. HCR amplifiers (Molecular Instruments) corresponding to each primary probe were aliquoted into separate PCR tubes and heated to 95 °C for 90 s, then placed into a light protected drawer until use. Heated amplifiers were diluted in amplification buffer and added to the samples, which were then incubated for 1 hr at room temperature in a humidified chamber. Samples were then washed five times for 10 min each in 5 X SSCT, once with PBS containing 1 µM Hoechst 33342 (Thermo Fisher Scientific, product number 62249), then stored at 37 °C in a humidified chamber in the dark until imaging.

The samples were imaged as previously described on an Andor Yokogawa CSU-X confocal spinning disc on a Nikon Ti Eclipse microscope at room temperature. The fluorescence signals were obtained using DAPI epifluorescence excitation and the 561 and 640 nm lasers (Coherent), and images were captured using a Prime 95B scMOS camera (Photometrics) with a 100 x objective (Plan Apo 100 x DIC, Nikon, oil, NA = 1.45, part number = MRD01905, pixel size: 0.09 µm). 8 µm Z stacks of Halo and FISH fluorescence were taken for puncta counting, with 1 µm steps between frames.

Analysis of ORF1-RNA colocalization was performed in 3D using python in the same way as ORF1 puncta colocalization above. The only additional consideration was to focus the analysis on cytoplasmic spots due to the high number of nuclear RNA spots (likely nascent transcripts) that we would not expect to localize to L1 RNPs. Spots were localized to either the cytoplasm or nucleus based on the nuclear signal. The mean intensity of the signal in the nuclear channel was determined for each spot. Then, a cutoff value for nuclear channel intensity was found to classify each spot as either nuclear or cytoplasmic using the average of 2 methods: k-means clustering (scikit-learn) and Otsu thresholding (skimage.filters.threshold_otsu). This calculation was performed separately for each ROI to account for cell-to-cell variation in nuclear channel signal. After classification, spots were displayed overlaid on the DAPI image to manually check that the classification was accurate. Detected spots classified as nuclear were removed from the analysis. When placing random spots, positions within the nucleus were avoided by using the predetermined thresholds: if a random position was chosen with corresponding nuclear signal greater than the threshold, then a new random position was chosen until an appropriate position was found. Spots were considered RNA-positive if the normalized RNA intensity exceeded 1.15. This threshold value was chosen based on normalized RNA intensities determined by spot-calling on the RNA channel of the colocalization images; the threshold value of 1.15 corresponds to the 3rd percentile of normalized RNA intensity for called RNA spots and the 89th percentile of normalized RNA intensity for random RNA spots.

## Retrotransposition assays and FACS

Retrotransposition assays were conducted as described previously using quasi-stable HeLa M2 cells maintaining a pCEP-puro L1RP element containing a GFP-AI cassette (*Mita et al., 2018*). Briefly, quasi-stable HeLa cells were seeded in three separate wells of a six-well plate (Fisher Scientific, product number 50-202-137) for each L1 construct using complete puromycin media with 1 µg/mL doxycycline. After 72 hours, the cells were washed once with DPBS and were stained with 1 µm Hoechst 33342 (Thermo Fisher Scientific, product number 62249) for 30 min at 37 °C. The cells were then lifted using 500 µL of TrypLE Express Enzyme (Gibco, product number 12604021), neutralized with 500 µL of DMEM, and pelleted in 1.5 mL microcentrifuge tubes. The cell pellets were resuspended in 300 µL of freshly prepared FACS buffer (DPBS + 0.5% FBS), filtered into round-bottom tubes with cell strainer caps (Fisherbrand, product number 352235), and stored on ice until run on the Sony SH800S flow cytometer. A total of 25,000 cells per condition were analyzed for GFP and Hoechst signals, which were compensated before analysis. The cutoff of GFP+ cells was set according to cells cultured for 72 h in complete puromycin media without doxycycline, and the average percentage of GFP+ cells across all three technical replicate wells was reported as a single biological replicate for a given L1 construct. The average GFP+ rates across three biological replicates for each L1 construct were plotted in Prism 9 (GraphPad) and were compared using a one-way ANOVA with Tukey's multiple comparison correction. Distributions of GFP intensity across cell populations were plotted using FlowJo (BD Biosciences).

For FACS analysis of cellular ORF1-Halo expression levels across ORF1p mutants, 2 million quasi-stable HeLa M2 cells maintaining a pCEP-puro L1RP element were seeded in a 10 cm tissue culture dish (Corning, product number 430167) in complete puromycin media. The following day, L1 expression was induced in the cells for 6 hr using 1 µg/mL doxycycline in the conditioned media. For the last hour of induction, the cells were stained with 100 nM HaloTag Ligand JF646 (Promega, product number GA1121) and 1 µm Hoechst 33342 (Thermo Fisher Scientific, product number 62249) at 37 °C. As above, the cells were then washed once with DPBS, lifted using TrypLE Express Enzyme, and pelleted in 15 mL conical tubes (Corning, product number 430052). The cell pellets were resuspended in 2 mL of FACS buffer, filtered into round-bottom tubes with cell strainer caps and stored on ice until run on the Sony SH800S flow cytometer. Far-red fluorescence intensity was analyzed for 50,000 cells per ORF1p variant. Distributions of Halo intensity across cell populations were plotted using FlowJo (BD Biosciences).

## Plasmid availability

All plasmids will be deposited in AddGene.

## Software availability

Code is available at https://github.com/liamholtlab/GEMspa (copy archived at *Holt, 2023a*) and https://github.com/liamholtlab/spot_detection (copy archived at *Holt, 2023b*).

## Acknowledgements

We thank Paolo Mita for piloting initial live-cell L1 imaging and for extensive guidance through the vast collection of L1 expression constructs. We thank Martin Taylor and David Giganti for guidance on the purification and handling of bacterially-expressed ORF1 protein. We thank members of the Holt lab and Boeke lab for insightful scientific discussion and critical reading of the manuscript. ORF1p cartoon representations were created with BioRender.com.

This work was funded by NIH R01 GM132447 and R37 CA240765, the Chan Zuckerberg Initiative, and the Air Force Office of Scientific Research (AFoSR) grant FA9550-21-1-3503 0091 to LJH and a subaward from NIH P01 AG051449 to JDB. SS was funded by the Vilcek MSTP Scholars Award and NIH Medical Scientist Research Service Awards T32GM007308 and T32GM136573.

## Additional information

### Competing interests

Jef D Boeke: is a Founder and Director of CDI Labs, Inc, a Founder of and consultant to Neochromosome, Inc, a Founder, SAB member of and consultant to ReOpen Diagnostics, LLC and serves or served on the Scientific Advisory Board of the following: Sangamo, Inc, Modern Meadow, Inc, Rome Therapeutics, Inc, Sample6, Inc, Tessera Therapeutics, Inc and the Wyss Institute. The other authors declare that no competing interests exist.

### Funding

| Funder | Grant reference number | Author |
|---|---|---|
| National Institute of General Medical Sciences | R01 GM132447 | Liam J Holt |
| National Cancer Institute | R37 CA240765 | Liam J Holt |
| Air Force Office of Scientific Research | FA9550-21-1-3503 0091 | Liam J Holt |
| National Institute on Aging | P01 AG051449 | Jef D Boeke |
| The Vilcek Foundation | Vilcek MSTP Scholars Award | Srinjoy Sil |
| NIH Medical Scientist Research Service Award | T32GM007308 | Srinjoy Sil Farida Ettefa |
| NIH Medical Scientist Research Service Award | T32GM136573 | Srinjoy Sil Farida Ettefa |

The funders had no role in study design, data collection and interpretation, or the decision to submit the work for publication.

### Author contributions

Srinjoy Sil, Conceptualization, Formal analysis, Investigation, Methodology, Writing - original draft, Writing – review and editing; Sarah Keegan, Data curation, Software, Formal analysis, Visualization, Methodology, Writing – review and editing; Farida Ettefa, Formal analysis, Investigation, Methodology, Writing – review and editing; Lance T Denes, Data curation, Formal analysis, Investigation, Methodology, Writing – review and editing; Jef D Boeke, Resources, Writing – review and editing; Liam J Holt, Conceptualization, Supervision, Funding acquisition, Writing - original draft, Project administration, Writing – review and editing

### Author ORCIDs

Srinjoy Sil ![ORCID] http://orcid.org/0000-0002-6722-4311
Jef D Boeke ![ORCID] http://orcid.org/0000-0001-5322-4946
Liam J Holt ![ORCID] http://orcid.org/0000-0002-4002-0861

### Decision letter and Author response

Decision letter https://doi.org/10.7554/eLife.82991.sa1
Author response https://doi.org/10.7554/eLife.82991.sa2

## Additional files

### Supplementary files
• MDAR checklist

### Data availability

All data generated or analysed during this study are included in the manuscript and supporting files.

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
