## [Editor Report]

This valuable study describes a new system for tracking the formation of puncta by ORF1p, a nucleic acid binding protein encoded by the L1 retrotransposon, *in vivo*. The fact that RNPs form "membrane-less" structures is already established in other situations as the authors point out, but the work provides better-defined biochemical features, especially for RNA association and *in vivo* dynamics. Overall, the evidence for the conclusions is solid, and the work will be of interest to colleagues studying retrotransposition as well as biomolecular condensates.

---

## [Decision Letter]

[Editors' note: this paper was reviewed by Review Commons.]

Thank you for submitting your article "Condensation of LINE-1 is required for retrotransposition" for consideration by eLife. Your article has been reviewed by 3 peer reviewers at Review Commons, and the evaluation at eLife has been overseen by a Reviewing Editor and Detlef Weigel as the Senior Editor.

Based on your manuscript, the reviews and your responses, we invite you to submit a revised version incorporating the revisions as outlined in your response to the reviews plus the following comments. Please take these concerns seriously, as the revised manuscript will be evaluated by experts among our Reviewing Editors.

When preparing your revisions, please address the following points in addition to your revision plan:

The narrative strongly claims that ORF1p undergoes "LLPS". This is a high bar to cross. What is the evidence to insist that the coexisting phases are in fact liquids? We presume you want to imply liquid to mean simple, Newtonian fluids. At a minimum, the prefix of "LL" must be dropped. "PS" is fine. However, the analysis focuses on RNA molecules influencing the material properties. You make a case for conversion to solids (an equilibrium phenomenon) but they also hint at a dynamical arrest behavior. Condensates are viscoelastic materials. The phase behavior is best described as a coupling of associative and segregative transitions. This is not the same as simple LLPS. And because of the complications of associative transitions, dynamical arrest of phase separation is a reality, especially with RNA molecules. The narrative unfortunately glosses over complexities in favor of a simple presentation, which becomes misleading. We agree that the in vitro reconstitutions are indeed incomplete. We would also offer that we do not know if arrested states are or can be processed by ATP dependent RNA binding proteins, i.e., helicases. The mutational work is interesting, but for now the material properties of the condensates are unresolved. Further, we do not know what material states are functionally relevant in cells. One should be careful with the term "LLPS" as soon as one sees a round object in diffraction limited microscopy. The internal organization of molecules will most certainly be inhomogeneous.

Therefore, please go beyond your current thinking about how you place condensates and condensation in the context of ORF1p functions. To help you, here are some recent papers that could aid in clarifying the nomenclature but also help in rethinking your experimental design and analysis.

- https://doi.org/10.1016/j.molcel.2022.05.018

- https://doi.org/10.1073/pnas.1821038116

- https://www.biorxiv.org/content/10.1101/2021.12.30.474545v2.

The experiments go beyond what has been published before, including work from the Furano, Williams and Weichenrieder labs, and it might be useful to discuss what methodological advances have allowed for the new insights described here.

---

## [Author Response]

General Statement

The goal of this study was to further our understanding of the behavior and function of the L1 ORF1 protein using live-cell imaging for the first time. We subsequently assayed the properties of purified wild-type and mutant ORF1p in vitro in order to determine the biochemical properties of the protein that enable formation of the homogeneous punctate assemblies that wild-type ORF1p forms in cells. Initially, we found discrepancies between the in vitro and cellular experiments, where the K3A/K4A and R261A ORF1p mutants form droplets in vitro but do not efficiently form puncta in cells. Upon assaying an additional set of ORF1p mutants, however, we found that measuring the variation of the physical properties of wild-type and mutant ORF1p droplets across RNA concentrations demonstrates that a high ORF1p sensitivity to RNA in vitro correlates with more efficient puncta formation in cells. This result along with experiments demonstrating *cis* preference of ORF1p in live cells are consistent with a model in which newly translated ORF1 proteins rapidly assemble with each other and their *cis* L1 RNA to form a stable L1 RNP. Importantly, we also present evidence that formation of ORF1p assemblies correlates with L1 retrotransposition activity.

We note that while Newton et al. (2021) published similar in vitro ORF1p droplet data, we performed ORF1p biochemistry experiments at much lower protein concentrations and included RNA to better approximate the ORF1p-RNA interactions involved in RNP assembly in cells. We additionally correlated the properties of these in vitro droplets with ORF1p puncta formation in cells. We acknowledge that the lack of a clear correlation between in vitro droplet formation and cellular puncta formation suggests that the cellular puncta likely do not form by conventional liquid-liquid phase separation (LLPS), and we are careful to suggest that the cellular puncta form through a kinetic and possibly cotranslational assembly or condensation mechanism rather than an equilibrium process like LLPS; this idea will be clarified in revisions to the main text. We also acknowledge that there are a number of loose ends that remain in need of further investigation, including a thorough analysis of nuclear ORF1p assemblies and a closer look at a hypothetical cotranslational ORF1p assembly mechanism. However, we believe that the work we have done in observing and analyzing the behavior of cellular ORF1p assemblies in live cells and correlating cellular behavior with retrotransposition activity and in vitro ORF1p droplet properties is a large step that merits publication before further characterization of specific aspects of cellular ORF1p.

Description of the planned revisionsReviewer #1Major comments:“On several occasions, the authors propose that ORF1p-HALO dynamics *in vivo* is linked to its co-translational association with L1 RNA. However, they never show the presence of L1 RNAs in ORF1p-HALO puncta *in vivo*. To strengthen the conclusion that the puncta observed *in vivo* are L1 RNPs, the authors should add experiments showing the presence of L1 RNA in the cytoplasmic puncta (by RNA FISH) or that the puncta are dependent on the presence of L1 RNA (expressing ORF1p-HALO alone should not be sufficient for puncta formation). These experiments seem to be realistic in few weeks with the tools already available in the laboratory.”

Showing the presence of L1 RNA in the condensates would certainly be helpful to cement the connection between our live-cell puncta observations and the retrotransposon life-cycle. We plan to design FISH probes against the HaloTag sequence in order to specifically visualize our modified L1RP RNA and optimize a protocol for visualizing both ORF1-Halo and the synthetic L1RP RNA. The experiment will be done on HeLa cells expressing L1 for 6 hours. We disagree that expressing ORF1-Halo by itself would definitely be insufficient for puncta formation, as we do not have data that would help us predict how the length or sequence of the ORF1p-encoding RNA will affect puncta formation in live cells. Additionally, observing the loss of puncta formation in this experiment could also indicate that ORF2p is necessary for puncta formation in cells.

We successfully completed the FISH experiments and conclude that *cis* L1 reporter RNA is indeed present within ORF1p-Halo condensates. The results of this experiment are in revised Figure 1G.

Minor comments:“Figure 1F: Having the pictures of cell nuclei (like in Figure 1D) would be nice to know how many cells we are looking at in this panel.”

We have added an outline of the single cell and nucleus in this panel.

“Figure 2E: it is surprising that there is no correlation between the ORF1p:RNA ratio and the number of individual fusion events (i.e. curves of ORF1p+RNA 10000:1 and 1000:1 overlap while 3000:1 is different). Could the authors discuss this point?”

Indeed, we mention in the text that it is interesting that the decrease in droplet fusion rate is non-monotonic with the amount of RNA added, with the intermediate amount of RNA leading to faster droplet fusion (similar to WT) and the lower and higher amounts of RNA resulting in slower fusion events. The fusion events in the 1,000:1 and 10,000:1 conditions tend to arrest as partial fusion intermediates and hence have a high aspect ratio plateau. As described later in the manuscript, the fusion time of Newtonian droplets is proportional to the ratio of the fluid’s viscosity to its surface tension and is additionally proportional to the size of the droplets undergoing fusion (*τ ≈ (η∕γ) · ℓ*; Eggers, Lister, and Stone 1999; Brangwynne, Mitchison, and Hyman 2011). We propose that the addition of 2-kb RNA to the ORF1 protein both increases the viscosity and modulates surface tension of the generated phase. It appears that increased droplet viscosity is the dominant effect at the low RNA concentrations, which is the predicted effect for long RNA molecules (>250 nt) based on simulated data (Tejedor et al. 2021). This accounts for the slow and incomplete fusions seen in the 10,000:1 condition. We posit that the faster fusion rates in the 3,000:1 condition is a result of increased surface tension of the droplets that is able to overcome the increase in droplet viscosity. At an even higher RNA concentration, it is likely that the accumulated negative charge of RNA on the surface of the droplets leads to decreased surface tension that subsequently leads to slow and incomplete fusions in the 1,000:1 condition. Increases in protein-nucleic acid stoichiometries that lead to increased surface tension at low nucleic acid concentrations and decreased surface tension at high nucleic acid concentrations with a fixed increase in viscosity have been described in in vitro studies using short peptides and DNA oligomers (Alshareedah, Thurston, and Banerjee 2021).

Interpretation of droplet fusion experiments was clarified in lines 328-331, 338-343, and 560-575.

Reviewer #2Major comments:“(A) The functional relevance of condensate formation by IDR-containing proteins has been questioned (Martin, E. W. and A. S. Holehouse – 2020; Emerging Topics in Life Sciences 4: 307). These authors conclude their review as follows: "In summary, IDRs are ubiquitous and play a wide range of functional roles across the full spectrum of biology, and in a large number (likely the majority) of cases their biological function has nothing to do with the ability to form large macroscopic liquid droplets. The notion that the presence of an IDR means a protein has evolved to phase separate is an inaccurate inference that has unfortunately been used to justify questionable lines of inquiry and questionable experimental design." And in terms of ORF1p this admonition is exemplified by the findings of Newton et al. (2021, Biophys J 120;2181) cited by the present authors. This study showed that phase separated condensates readily form by just the N-terminal 152 amino acids (NTD + coiled coil). As this region of ORF1p cannot bind NA, condensate formation is indifferent to RNA binding, an obviously critical function of ORF1p.”

We certainly agree that IDRs are ubiquitous and are in many cases affinity-tuning domains. As binding affinity is a key biochemical parameter, IDRs are likely generally crucial for biological function. important for biological function. In the case of ORF1p, a protein with no catalytic activity whose primary function is an RNA chaperone that acts in a unique cis-preferential manner, we argue that IDR-mediated condensation of ORF1p with RNA plays a significant role in the protein’s function. We agree that the sufficiency of the N-terminal 152 amino acids of ORF1p for droplet formation is intriguing because of its independence on RNA binding; this is why we pursued the study of full-length wild-type ORF1p and RNA-binding mutant ORF1p R261A in vitro, with and without RNA, and in cells. We do not agree that the fact that the N-terminal half of ORF1p can undergo phase separation without RNA binding indicates that phase separation is unimportant for ORF1p function, as we have shown that in vitro ORF1p condensate properties (e.g. inverse capillary velocity) change with the addition of RNA and that ORF1p R261A fails to form punctate assemblies in cells. Thus, the ability to form higher order assemblies in a simple reconstitution cannot predict assembly states in a complex intracellular environment. By analogy, kinases may phosphorylate targets in vitro, but there may be low to undetectable phosphorylation at steady-state *in vivo* due to the opposing action of phosphatases. This is why detailed *in vivo* experiments are crucial. We would argue that the Newton et al study is an interesting reductionist approach that provides structural insights, while our results indicate that *in vivo* L1 RNP assemblies involve a more complex network of interactions, within which RNA-ORF1 interactions are crucial.

The similar but distinct roles of K3/K4 and R261 in our in vitro ORF1p condensation experiments have been clarified in lines 387-405 and Figure 3 Supp 3.

“(B) Earlier studies (Ostertag et al. – 2000; NAR 28:1418) showed that sufficient retrotransposition events have occurred by 48 hours after introduction of an L1 retrotransposition reporter to be readily detectable by whole cell staining for the retrotransposition-generated reporter gene product. The 48-hour lag presumably reflects the time to accumulate sufficient L1RNPs or their retrotransposed products to be detectable. Does this mean that the puncta (Figure 1F) accumulating during the first 24 hours after introduction of their full-length L1 retrotransposition reporter (Figure 1C) are the L1RNPs generated by the reporter? If not, what are they? If they are L1RNPs, are they thought to be or expected to exhibit the properties of phase separated condensates or are such properties just a feature of disembodied ORF1p that the authors posit could form an active L1RNP? The Ostertag paper should be cited here given its relevance to this issue.”

Yes, we believe that the puncta accumulating in the first 24 hours (and even in the first 6 hours) of expression of the full-length L1 reporter are the L1 RNPs generated by the reporter. We note that the expressed ORF1p localizes to discrete bright foci (dense phase) against a background of dim cytoplasmic background (dilute phase); this is a hallmark of condensation. We do not believe that these assemblies are liquid-like given their uniformity in size and the dearth of fusion events observed in live-cell imaging, which is particularly exemplified in the two-color *cis* preference experiments. However, the assemblies could still be liquids with low surface tension that do not undergo fusion (Alshareedah, Thurston, and Banerjee 2021), or they could be gel-like assemblies (Maharana et al. 2018). We posit that the liquid-like behavior of purified ORF1p in vitro may be representative of ORF1p behavior during L1 RNP assembly, but the assembled, diffusive foci we primarily observe do not readily undergo fusion events (Discussion paragraph 2). Notably, the addition of varying amounts of RNA to the in vitro ORF1p condensates also decreases their propensity to undergo fusion events, which is more similar to the behavior of the L1 RNPs in cells than the behavior of in vitro condensates with ORF1p alone (Discussion paragraph 3). To clarify the point about the timing of retrotransposition, we believe that a sufficient number of L1 RNPs for retrotransposition are likely generated in less than 24 hours but that other steps, including nuclear translocation, target site identification, successful TPRT, and accumulation of GFP from the newly integrated reporter, require the remainder of the 48 hours. We will incorporate these ideas and our expectations into the first section of the Results as well as the Discussion, and Ostertag et al. 2000 will be cited appropriately. Additionally, using RNA FISH to detect our modified L1RP RNA inside the ORF1p puncta as described previously will confirm that these assemblies that are formed after 6 hours are likely to be L1 RNPs.

We successfully completed the FISH experiments and conclude that *cis* L1 reporter RNA is indeed present within ORF1p-Halo condensates after only 6 hours of expression induction. The results of this experiment are in revised Figure 1G.

“(C) Four of the IDRs in ORF1p harbor or are juxtaposed to phosphorylation sites essential for retrotransposition (their citation – Cook et al., 2015). As the authors expressed their purified proteins in *E. coli*, it is not phosphorylated and would not only be inactive for retrotransposition and given the structural effects of phosphorylation (e.g., Bah, A., et al.;2015; Nature, 510, 106) it would differ significantly from the structure of the active protein. As variables they introduce into ORF1p several not too subtle mutations particularly regarding the ORF1 coiled coil. They thereby aim to assess the role or particulars of ORF1p condensate formation for L1 retrotransposition. In their Abstract they state (p.1, l. 11) "…we propose that ORF1p oligomerization on L1 RNA drives the formation of a dynamic L1 condensate that is essential for retrotransposition."”

Yes, we are aware of Cook et al. (2015) and their data supporting the necessity of ORF1p phosphorylation by PDPKs for L1 retrotransposition in cells. As you point out, our purified ORF1 protein was expressed in *E. coli* and is therefore not phosphorylated, and phosphorylation of ORF1p, particularly in the NTR (S18 and S27), may change the propensity of ORF1p to undergo condensation. However, phosphorylation in cells is a dynamic process that likely links ORF1p phosphorylation to host cell behavior, like cell cycle state and stress response. In Cook et al. (2015), the phosphomimetic mutants S18D and S27D largely rescued the retrotransposition deficits incurred by S18A and S27A mutations, respectively, but S18D and S27D notably still had reduced retrotransposition rates compared to wild-type ORF1p, with the double phosphomimetic mutant S18D/S27D having less than 50% of wild-type activity. As a result, we did not feel that characterizing a “fully phosphorylated” ORF1p in vitro would be inherently more representative of ORF1p condensation behavior, even though comparing the droplet formation of phosphorylated and unphosphorylated ORF1p in vitro could certainly be interesting. One phosphomimetic mutant (S27D) was studied in Newton et al. (2021), where they found that the mutant may have enhanced droplet formation compared to wild-type. It will be interesting to further explore phosphorylation in the future.

“(D) Although the authors provide no direct experimental evidence for the above statement and whatever the authors mean by "dynamic L1 condensate" how does this conclusion materially differ from the conclusions published by Naufer et al., in 2016 (NAR; 44,281), which also was not cited by the authors. Naufer et al. used single molecule studies and highly purified ORF1p that had been expressed in insect cells (and thus was fully phosphorylated, Cook et al., 2015). They showed that oligomerization of nucleic acid (NA)-ORF1p complexes to a compacted stably bound polymer was positively correlated with retrotransposition. Both properties could be eliminated by coiled coil mutations that had no effect on biochemical assays of ORF1p activity – high affinity NA binding and NA chaperone activity. As both properties map to the carboxy terminal-half of ORF1p, the inactivating coiled coil mutations are an example of the numerous instances of strong epistasis exerted by amino acid substitutions in the coiled coil on the retrotransposition activity of ORF1p. In some cases epistasis is exerted at the single residue level (e.g., Martin,et al. – 2008, Nucleic Acids Res., 36, 5845; Furano, et al. – 2020, PLOS Genetics 16 e1008991.)While the authors are apparently also not mindful of the PLOS Genetics paper examining the effect of a single inactivating coiled coil substitution at the level of microscopically observed condensates could have provided compelling evidence linking their formation and retrotransposition. On the other hand, lack of a condensate-based readout for single amino acid inactivating coiled coil mutations would question the validity of equating ORF1p condensates with retrotransposition competence.(E) The afore mentioned Callahan et al. study (2012, NAR, 40, 813) in addition to producing results partly recapitulated in Figure 2 of the present paper, showed that ORF1p polymerization was mediated by interactions between the highly conserved RRM-containing region of ORF1p. This observation is consistent with previous studies showing RRM-mediated protein interactions of other proteins (Clery, et al. 2008, Curr. Opin. Struct. Biol., 18, 290; Kielkopf, et al. Genes Dev., 18, 1513)”

We agree, it was an oversight to miss citing Naufer et al. (2016), as our results are a logical extension of their findings that indicate that ORF1p function can be lost with alterations in ORF1p oligomerization kinetics on nucleic acids (NAs), even if NA binding affinity and NA chaperone activity is preserved. Our assertion in Figure 4 is that distinct alterations of the physical properties (inverse capillary velocity) of ORF1p condensates in response to changing RNA concentrations is predictive of both punctate assembly of ORF1p in cells and retrotransposition activity. Indeed, this conclusion is similar to that of Naufer et al., as ORF1p oligomerization kinetics likely dictate the physical properties of the resulting ORF1p-RNA condensates. Importantly, our work shows that changes in these ORF1p assembly properties that are measured in vitro result in changes in L1 RNP formation in cells as measured by live-cell confocal microscopy, which in turn correlate with retrotransposition rate. Callahan et al. (2012) similarly finds that full-length ORF1p is able to stabilize or “cage” mismatched duplex DNA at low protein concentrations but melt it at higher protein concentrations, while the predominantly monomeric RRM fragment of ORF1p melts the mismatched duplex DNA at low and high protein concentrations without exhibiting a caging effect, thus arguing that trimer or higher-order oligomer structure (rather than RRM properties alone) is critical for particular ORF1p-NA interaction characteristics that are necessary for retrotransposition. Our findings are in accordance with this conclusion, as we find that the physical properties of ORF1p-RNA condensates in vitro are altered by mutations in the NTR, CC, and RRM of ORF1p in ways that correlate with assembly formation in cells and retrotransposition. While Callahan et al. provides evidence that the RRM may be able to oligomerize without the coiled coil, the assertion is complicated by the retention of 3.5 heptads of the coiled coil in RRM construct, including two chloride ion-coordinating heptads and a canonical stabilizing salt bridge that likely contribute to oligomerization even in the absence of the remainder of the coiled coil (Khazina and Weichenrieder, 2018). Their observation that the transition from full-length ORF1p-mediated mismatched duplex DNA caging to DNA melting is dependent on ORF1p concentration rather than ORF1p:NA stoichiometry may reflect that this transition is dependent on ORF1p condensation, which occurs at a particular critical protein concentration and is independent of NA concentration (Alberti et al. 2019). Both Callahan et al. and Naufer et al. hypothesize that ORF1p-NA interaction properties may be able to drive cotranslational assembly of ORF1p on its *cis* RNA, thus allowing for *cis* preference; our work is in agreement with this notion and provides the first microscopy-based evidence of the *cis* preference of two species of ORF1p in L1 RNP assemblies that are co-expressed in the same cell. This discussion of the intersection between our work and Callahan et al. and Naufer et al. will be adapted and added to the Discussion section of our manuscript.

Minor comments:“(1) p.2. l. 8, the citation to TPRT should include Luan,et al.- 1993, Cell 72: 595”

This citation was added in line 76.

“(2) p. 5, middle of 2nd para – what does "different diffusivity" mean?- what are "stereotyped puncta"?”

“Different diffusivity” refers to the starkly decreased effective diffusion (D_eff_) of nuclear ORF1 puncta compared to cytoplasmic puncta (Figure 1D, Figure 1-Supp 1D, Supp Movie 1). “Stereotyped puncta” refers to the uniform and homogenous nature of the ORF1 puncta in terms of their size and shape. These terms will be replaced for clarity.

These terms were replaced in lines 185 and 188.

“Any invocation of cis preference should cite the foundational study by Kaplan, N., et al. (1985). "Evolution and extinction of transposable elements in Mendelian populations." Genetics 109 459.”

This citation was added in line 194.

“(3) p.10 middle paragraph, the authors state:"The decreased phase separation of the R261A mutant was unexpected, as we predicted that mutating a core RNA-binding residue would only affect condensation in the presence of RNA. We also noted that the protein partition coefficients of the R261A condensed phases were higher than their counterparts for WT and K3A/K4A. Taken together, these experiments showed that K3/K4 and R261 are not essential for protein condensation in vitro."These findings would have been predicted by the afore mentioned findings of Newton et al., which should be cited here.”

Newton et al. find that “a minimal construct containing the N-terminus and coiled-coil domain (ORF1_1-152_) phase separates readily, similar to the full-length protein.” This finding predicts that a single-residue mutation in the RRM would be unlikely to affect droplet formation, which contrasts with our data for R261A. However, Newton et al. does have data indicating that neither K3/K4 or R261 are essential for protein condensation and will be cited appropriately here.

This citation was added in line 381.

“(4) p. 14, first paragraph "we predicted that stammer-deleted ORF1p would maintain an elongated coiled coil conformation that might disfavor trimer- trimer interactions that are mediated by the N terminal half of the protein (Figure 4A, left two cartoons)."It seems that the authors are stating that different fully formed trimers can form larger complexes mediated by interactions between their coiled coils, an idea apparently based on results published by Khazina and Weichenreider (2018). This paper states that "Additional biophysical characterizations suggest that L1ORF1p trimers form a semi-stable structure that can partially open up, indicating how trimers could form larger assemblies of L1ORF1p on LINE-1 RNA." However, the cited Khazina structural data ((PDB) entry (6FIA)) were derived from coiled coils that had been solubilized to monomers in guanidinium HCl from inclusion bodies (insoluble aggregates) that had accumulated during their synthesis in *E. coli*…a common condition for highly expressed proteins. Fully denatured ORF1p coiled coils such as these, which also lack the entire NTD are an in vitro artifact and never exist in "nature". It is almost certain that ORF1p monomers trimerize while being synthesized on adjacent ribosomes (e.g., Bertolini et al.- 2021; Science 371: 57). I am not aware of any biochemical evidence from the Martin laboratory on mouse ORF1p or the Weichenrieder or Furano laboratories on human ORF1p indicating that the coiled coils of fully formed trimers synthesized *in vivo* can unravel to mediate interactions between different trimers. In fact, the authors' results in Figure 1F supports this contention.”

Despite the purification of their protein from inclusion bodies, we believe that the non-canonical heptads in the N-terminal half of the ORF1p coiled coil and the stammer-induced overwinding likely confer an increased propensity for the trimeric coiled coil to adopt a more open conformation that exposes normally buried a- and d-layer residues, as proposed in Khazina and Weichenrieder (2018). However, no data thus far has identified specific inter-trimer coiled-coil interactions that would drive polymerization. Newton et al. suggest that interactions between residues 47-53 and 132-152 are important for phase separation in vitro; these interactions may be affected by a decrease in coiled coil opening as a result of stammer deletion. Interactions between trimers are necessary for condensate formation, and therefore the observation of ORF1p puncta formation in cells (as in Figure 1F) would support the importance of inter-trimer interactions. We believe we are working with the best model for the existing data. However, we have edited the language in this section to highlight the hypothetical nature.

“(5) p.10, Legend to Figure 1GThe cells were stained simultaneously with two Halo ligand dyes (Halo-JF549 and Halo- JF646), giving a positive control for colocalization.Why is staining the same ligand (Halo) with two different dyes a colocalization control?”

We believe that having an internal positive colocalization control in each analyzed field of view better demonstrates the lack of colocalization between the two types of tagged ORF1 proteins and validates the technical sensitivity of this type of colocalization analysis. Since each ORF1p puncta likely contains numerous ORF1p trimers, simultaneously staining the same Halo tag with two ligand colors should in theory equally co-stain all of the ORF1p-Halo foci and have a colocalization of 1.0. Due to imperfections with co-staining, imaging, and colocalization analysis, the average co-stained Halo colocalization was around 0.5, but we felt that including that data was helpful to contextualize the corresponding result that the average colocalization between ORF1-Halo and ORF1-mNG2 was close to 0.

A more sophisticated analysis was applied to this dataset, allowing for quantification of mNG2 intensity and JF646 intensity at JF549+ spots versus random intracellular spots. See lines 214-222 and Figure 1F.

“(6) The authors conclude their paper with the statement "The L1 system characterized in this work employs a uniquely powerful combination of biochemical reconstitution, live-cell imaging, and functional phenotyping in cells. in vitro reconstitution allows us to study the biophysical properties of condensates in a minimal and controllable system."However, there are several instances where the in vitro biochemical properties of ORF1p variants are somewhat discordant with their *in vivo* results. In the case of their coiled coil mutants. replacement of the coiled coil stammer, MEL (uniquely invariant for more than 50 Myr of primate coiled coil evolution) with AAA or AEA exhibited reduced retrotransposition that was not accompanied by a corresponding reduction in condensate formation (Figure 4). In another instance, while mutation of the highly conserved residue (R261) necessary for RNA binding eliminated retrotransposition it did not have a corresponding effect on condensate formation even in the presence of RNA (Figure 3).”

Indeed, translating the in vitro condensation experiments to understand the cellular condensation behavior of ORF1p was not straightforward. As you point out, the R261A mutant was a clear example that the formation of in vitro protein droplets in our assay was not directly predictive of cellular assembly. For this reason, we saught correlations in the more detailed physical properties of the in vitro ORF1p droplets and noted that the inverse capillary velocity of the in vitro condensed phases of the WT and mutant ORF1 proteins had varied responses to the addition of RNA (Figure 4E). The WT and StammerAEA ORF1 proteins, which both condense in cells, had a stark increase in inverse capillary velocity in response to low amounts of RNA (10,000:1 RNA) while K3A/K4A and R261A had no response and a gradual response, respectively. This indicated to us that the physical material properties of the in vitro ORF1p condensed phases, rather than the extent to which they formed droplets, are indicative of the ability of ORF1p mutants to undergo assembly in cells. This idea will be clarified and included in the Discussion section.

The discussion of the material properties of in vitro ORF1p condensates was clarified in lines 560-575 for this discussion. Additional nanomolar protein concentration characterization and analysis was performed to better associate our in vitro findings with the observed cellular ORF1 puncta (Figure 4E, Figure 4 Supp 5; lines 577-598).

“General comments on the Figures – Although I rather liked the cartoon version of ORF1p (Figure 1B) and when used to show the location of mutated site, versions that purport to show the effect of mutations on structure (Figure 4A) are misleading and should be eliminated.”

We will make it clear that the cartoon representations of all ORF1p conformations are hypothetical.

Reviewer #3Major comments:“1. What is the evidence that ORF1p forms condensates in an endogenous situation? A more thorough discussion of the evidence, based on the literature is needed. Alternatively, authors could use antibodies (if available) to demonstrate that such structures indeed exist in cell culture of tissues.”

Since L1 expression is rather low in most normal somatic tissues, little characterization of endogenous ORF1p has been done. L1 upregulation in the context of oncogenic transformation has been demonstrated in a broad array of tumor samples across a number of studies (Rodic et al. 2014), but this detection of ORF1p was largely done using IHC and low-magnification imaging. The primary characterization of endogenous ORF1p in cytoplasmic aggregates was done in NTERA-2 and 2102EP cells, both of which are embryonal carcinoma cell lines (Goodier et al. 2007 – Figure 3C,E and Figure 5G,J-L; and Pereira et al. 2018 – Figure 1A,C,E, Figure 2A, and Figure 3D,E). Other cytosolic assembly formation has been shown using L1 expression constructs with untagged ORF1p (Sharma et al. 2016 – Figure 1; and Adney et al. 2018 – Figure 6). Discussion of this previous work will be added to the Results section of the manuscript.

See lines 171-182 for this discussion.

“2. The model that the observed puncta form co-translationally through co-condensation of ORF1p and its encoding mRNA is intriguing and would indeed provide an elegant biophysical explanation for the discussed cis preference of transposition. In my opinion, this idea is the strongest part of the paper. I would advise the authors to provide more compelling evidence for this idea, as currently, it is not well-supported by the data. At the least, the authors need to show that the L1 mRNA is actually present in the studied condensates (for example, using smFISH on fixed cells). This will also allow the determination of the number of L1 mRNAs present in each condensate.”

Yes, we plan to optimize a protocol for RNA FISH of our inducible L1RP RNA to visualize and quantify L1RP RNA in cellular ORF1p assemblies as described above.

This comment is addressed in the revised Figure 1G in which we detected L1 RNA in ORF1-HaloTag condensates by FISH.

“4. Please provide a more detailed analysis of the formation of nuclear ORF1p condensates. How much later do they appear? The nucleus is the place where transposition occurs. Do the authors suggest that the co-translationally formed condensates enter the nucleus? Or do they form there de-novo? Is there also no colocalization in the nuclear foci? This could be addressed by a quantitative time-course.”

We agree that this is a very interesting question, but we also acknowledge that a full characterization of nuclear ORF1p puncta, their behavior across the cell cycle, and their relationship with retrotransposition would be a significant amount of work that would likely merit its own manuscript. We have preliminary evidence that the co-translationally-formed ORF1p puncta enter the nucleus during nuclear envelope breakdown in mitosis rather than forming de novo.

This analysis will be part of a subsequent paper.

“5. The in vitro assays only use the L1 mRNA fragment. Do other RNAs (for example total RNA, rRNA, mRNA) similarly affect ORF1p condensates? Other studies showed that the presence of specific RNA could nucleate the formation of condensates in vitro, particularly where non-specific RNA is also present, mimicking the cellular environment (Maharana et al. Science 2018, PMID: 29650702; Elguindy and Mendell, Nature 2021, PMID: 34108682). The authors should test if the observed effect of L1 mRNA fragment is sequence-specific. Length dependence should also be addressed, as it may be the key parameter for the "co-translational assembly and gelation" model.”

We agree that exploring the effect of RNA characteristics on protein-RNA co-condensate behavior would be interesting. Preliminary data with a 2-kb control luciferase RNA (similar length to the L1 RNA we used) showed little effect on co-condensate formation. However, droplet fusion analysis was not done to evaluate the physical properties of the condensates formed by the two RNA species.

This analysis will be part of a subsequent paper.

Minor comments:“1. The K3/K4 and R261 variants don't form puncta and do not promote transposition, yet phase separate at a similar concentration in vitro. The stammer mutants phase separate less efficiently in vitro, yet form puncta and promote transposition. This suggests that the in vitro phase separation assay is not very informative of the protein's behavior in cells. To me, it suggests that the puncta observed in cells might not be formed through phase separation. Other mechanisms of puncta formation should be explored.”

We agree that the simple process of phase separation we observe in vitro does not directly correlate with the behavior of the assemblies observed in cells. We also use the term condensation, not phase separation to describe the assembly of the cellular puncta. As you point out, the R261A mutant was a clear example that the formation of in vitro protein droplets in our assay (which was modeled to be more physiologic using relatively low protein concentrations and adding RNA) was not directly predictive of cellular assembly. For this reason, we analyzed the physical properties of the in vitro ORF1p droplets more closely and noted that the inverse capillary velocity of the in vitro condensed phases of the WT and mutant ORF1 proteins had varied responses to the addition of RNA (Figure 4E). The WT and StammerAEA ORF1 proteins, which both condense in cells, had a stark increase in inverse capillary velocity in response to low amounts of RNA (10,000:1 RNA) while K3A/K4A and R261A had no response and a gradual response, respectively. This pattern indicated to us that the physical properties of the in vitro ORF1p condensed phases, rather than the extent to which they formed droplets, correlate with the ability of ORF1p mutants to undergo assembly in cells. We agree that this discrepancy between in vitro and cellular assembly suggests that ORF1p assembly does not occur solely through an equilibrium demixing process like phase separation and likely involves a kinetic, possibly cotranslational, component that is difficult to model in vitro. This idea will be clarified and included in the Discussion section.

This comment is addressed in part with the revised Figure 4E and Figure 4 Supp 5, in which lower concentrations of ORF1 protein in vitro yielded punctate condensates that resemble those seen in cells expressing our L1 construct. Further acknowledgment and discussion of dynamical arrest rather than LLPS can found in lines 168-171 and 633-643.

“6. It would be great to see how the StammerDel behaves in vitro. The authors could at least try the purification with their current protocol. Full-length proteins often behave very differently than the fragments alone.”

We plan to do an expression test of the StammerDel ORF1p to determine if it ends up in inclusion bodies and if it remains soluble using our current purification protocol.

ORF1p StammerDel was found to be insoluble following bacterial lysis and pelleted as an aggregate during clearing spins.

Description of the revisions that have already been incorporated in the transferred manuscriptDescription of analyses that authors prefer not to carry out.Reviewer #3Major comments:“3. If authors have access to a microscope that can perform FRAP measurements, I would strongly suggest such an assay, where the individual cytoplasmic and nuclear ORF1p puncta can be examined for their material properties as a function of time (compare 6 hours post-induction and 72 hours post-induction).”

The cytoplasmic puncta are highly mobile at the 100-ms-timescale, and therefore it would be difficult to track their fluorescence recovery; however, it may be possible to get a small fraction of puncta that remain stationary enough to adequately assess for FRAP. Nuclear puncta might be possible to FRAP due to their lower effective diffusion, but they are barely above the defraction limit, making FRAP extremely difficult. Furthermore, they may not exhibit FRAP due to the markedly lower concentration of ORF1p in the nucleus compared with the cytoplasm (as opposed to not exhibiting FRAP due to truly low protein exchange with the surrounding solution). Based on our *cis*-preference experiments, however, we would predict that the cytoplasmic ORF1p puncta would be unlikely to exhibit FRAP, since if they were rapidly exchanging protein, the puncta would start out single-colored but would eventually equilibrate to dual-colored due to stochastic incorporation of ORF1 proteins with either tag. Hence, our model is that protein-RNA condensation plays a role in assembly of L1 RNPs but that the RNPs do not remain liquid-like following assembly.

Minor comments:“2. Based on the fluorescent images, can the authors estimate what percent of the ORF1p protein is actually present in distinct condensates and how much is diffuse in the cytoplasm or nucleoplasm? How does the outside (diffuse) concentration change upon increased expression or ORF1p? Is there any evidence of a saturation concentration?”

We measured the fluorescence intensities of ORF1 puncta at 6 and 24 hours of dox induction and found that there was only a slight increase in puncta intensity with the significantly increased protein expression (Figure 1, Supp 1B). However, there certainly are cells with almost undetectable diffuse ORF1p signal and a few ORF1 puncta as well as higher-expressing cells that have many puncta but high concentrations of diffuse ORF1p, which argues against a clear saturation concentration. Estimating the fraction of ORF1 protein in condensates vs diffuse would be difficult given the lack of robust and unbiased method for segmenting the dynamic ORF1 puncta against moderate-to-high levels of diffuse ORF1p fluorescence. Since we are not arguing that the assemblies in cells are forming via equilibrium phase separation, we do not feel that this complex fluorescence microscopy analysis would yield highly informative results, as we would predict that there is not a clear saturation concentration.

“3. Does the ORF1-Halo and ORF1-mNG2 colocalization change at longer time-points where larger condensates are observed?”

We have not observed the two species of ORF1p at longer time-points. While this would be a relatively simple experiment, we do not believe that the result would be informative since the shorter induction generates puncta that are similar to those seen by immunofluorescence of endogenous ORF1 foci in embryonal carcinoma cell lines, while the longer induction forms more stress-granule-like assemblies. Interpretation of experiments with longer induction times are also complicated by cells undergoing cell division, which is likely when L1 RNPs interact with chromatin and enter the nucleus; since the details of nuclear entry and RNP-chromatin interactions are not known, we do not have a clear expectation for how that process would affect colocalization.

“4. The authors often refer to the "the total area of condensed phase". This parameter is not very useful, as it highly depends on the experimental condition. Instead, authors should determine the apparent saturation concentration for each studied mutant in the presence and absence of RNA at a relevant RNA concentration. This requires increasing the resolution at the protein concentration axis and an unbiased analysis pipeline.”

While this is a fair point, we do not believe that performing droplet assays with high protein concentration resolution with and without RNA across all five ORF1p variants would be an efficient use of revision time; we have clear evidence that the propensity of the ORF1p variants to form droplets does not prefectly correlate with cellular assembly and retrotranspition. Rather, the physical properties of the droplets predict *in vivo* activity. Therefore, determining saturation concentrations for each variant is not of great interest. We feel that it is more important to focus on revision experiments pertaining to the behavior of the cellular assemblies and the physical properties of the in vitro droplets.

“5. It is shown that decreasing ORF1p protein concentration at a fixed salt concentration decreased the total condensed phase area but increased the protein partition coefficient. The DNA/RNA binding mutant R261A does not show this trend. Moreover, it is the only mutant that shows a change in the phase diagram upon the addition of RNA. One explanation is that there are nucleic acid contaminants present in the protein prep. In fact, the R261A mutant seems to also have a lower 260 nm peak relative to 280nm peak at the chromatogram. That the enrichment of the ORF-1p protein changes with increasing concentration strongly suggests that we are already looking at a multi-component system here, where the contaminant would be a second component. The authors do include an extra step in the purification protocol to reduce nucleic acid contamination. However, they could also run an ion-exchange chromatography to improve the purity. Alternatively, they could test if adding benzonase, RNAse or DNAse changes the phase diagram of the ORF1p alone.”

As you mention, we were cognizant of the possibility of nucleic acid contaminants from our protein preps interfering with our protein-RNA condensation experiments and attempted to maximize their removal with an extended RNase and benzonase treatment. In fact, we used far more extensive nuclease treatments than previous studies (see methods). We were reassured that the carryover nucleic acid was likely not interfering with our experiments when we observed stark changes in the droplet fusion behaviors of WT ORF1p condensates even with the lowest concentration of RNA (10,000:1 RNA; Figure 2E). The data shown in Figure 4, Supp 3A exemplifies the behavior of all of the mutants over a greater range of RNA concentrations; notably WT and R261A are both less sensitive to higher concentrations of RNA than the other mutants (in that neither undergo a transition to more fibrillar structures at 1,000:1 RNA), while K3A/K4A, which appears to have the highest nucleic acid carryover, is among the most sensitive to higher concentrations of RNA.

[Editors’ note: further revisions were suggested prior to acceptance, as described below.]

When preparing your revisions, please address the following points in addition to your revision plan:The narrative strongly claims that ORF1p undergoes "LLPS". This is a high bar to cross. What is the evidence to insist that the coexisting phases are in fact liquids? We presume you want to imply liquid to mean simple, Newtonian fluids. At a minimum, the prefix of "LL" must be dropped. "PS" is fine. However, the analysis focuses on RNA molecules influencing the material properties. You make a case for conversion to solids (an equilibrium phenomenon) but they also hint at a dynamical arrest behavior. Condensates are viscoelastic materials. The phase behavior is best described as a coupling of associative and segregative transitions. This is not the same as simple LLPS. And because of the complications of associative transitions, dynamical arrest of phase separation is a reality, especially with RNA molecules. The narrative unfortunately glosses over complexities in favor of a simple presentation, which becomes misleading. We agree that the in vitro reconstitutions are indeed incomplete. We would also offer that we do not know if arrested states are or can be processed by ATP dependent RNA binding proteins, i.e., helicases. The mutational work is interesting, but for now the material properties of the condensates are unresolved. Further, we do not know what material states are functionally relevant in cells. One should be careful with the term "LLPS" as soon as one sees a round object in diffraction limited microscopy. The internal organization of molecules will most certainly be inhomogeneous.

We thank the reviewer for these comments. We agree that the process underlying ORF1p condensation is not simple LLPS and have updated the manuscript throughout to be more rigorous with our language. These edits are extensive, but some major examples to highlight are found on lines 96-114, 168-171, and 633-643 of the updated manuscript. Additional biochemical characterization was performed at nanomolar protein concentrations to help bridge the gap between our cellular findings and *in vitro* condensation results (Figure 4E, Figure 4 Supp 5; lines 577-598). Additional analysis of the *in vitro* experiments was performed and included to better support our interpretations of the roles of ORF1 mutations on condensate properties (Figure 2 Supp 2, Figure 3 Supp 2 and 3, Figure 4 Supp 3). RNA FISH against our L1 reporter RNA was performed with concurrent HaloTag labeling to confirm the presence of *cis* RNA in intracellular ORF1 puncta, with updated, more robust spot detection and quantitation methods used to detect ORF1p-RNA colocalization and assess for *cis* preference in ORF1p puncta (Figure 1F-G).